# EARL: Towards a Unified Analysis-Guided Reinforcement Learning Framework for Egocentric Interaction Reasoning and Pixel Grounding

Yuejiao Su [* 1]   Xinshen Zhang [* 1]   Zhen Ye [2]   Lei Yao [1]   Lap-Pui Chau [1]   Yi Wang [† 1]

https://github.com/yuggiehk/EARL

## Abstract

Understanding human–environment interactions from egocentric vision is essential for assistive robotics and embodied intelligent agents, yet existing multimodal large language models (MLLMs) still struggle with accurate interaction reasoning and fine-grained pixel grounding. To this end, this paper introduces **EARL**, an **E**gocentric **A**nalysis-guided **R**einforcement **L**earning framework that explicitly transfers coarse interaction semantics to query-oriented answering and grounding. Specifically, EARL adopts a two-stage parsing framework including coarse-grained interpretation and fine-grained response. The first stage holistically interprets egocentric interactions and generates a structured textual description. The second stage produces the textual answer and pixel-level mask in response to the user query. To bridge the two stages, we extract a global interaction descriptor as a semantic prior, which is integrated via a novel Analysis-guided Feature Synthesizer (AFS) for query-oriented reasoning. To optimize heterogeneous outputs, including textual answers, bounding boxes, and grounding masks, we design a multi-faceted reward function and train the response stage with GRPO. Experiments on Ego-IRGBench show that EARL achieves 65.48% cIoU for pixel grounding, outperforming previous RL-based methods by 8.37%, while OOD grounding results on EgoHOS indicate strong transferability to unseen egocentric grounding scenarios.

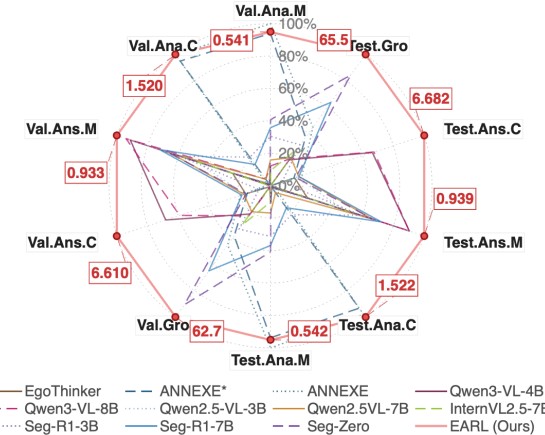

*Figure 1.* **Our method versus other MLLMs** on the Ego-IRGBench dataset. All scores are normalized to $[0, 1]$ for comparative visualization.

## 1. Introduction

Exploring human interactions with the environment or objects via visual data has consistently been a central concern in computer vision (Jahangard et al., 2024; Qian & Fouhey, 2023; Xu et al., 2023). Traditionally, studies on human-environment interaction have primarily concentrated on the third-person view (TPV, or exocentric) (Li et al., 2025b; Do & Kim, 2024; Lei et al., 2024). In recent years, the widespread adoption of head-mounted devices (e.g., Go-Pro) (Luo et al., 2024) and the continuous emergence of egocentric (or first-person view, FPV) benchmark datasets (Hummel et al., 2024; Dang et al., 2025; Fan et al., 2024) have increasingly shifted research attention toward the first-person perspective. Compared to exocentric data, egocentric views capture dynamic human-environment interactions in a more natural and immersive manner, reflecting not only the wearer's immediate experience but also their underlying intentions and goals to some extent (Yang et al., 2024; Zhou et al., 2025). Consequently, a growing number of researchers are investigating human-environment interactions from the egocentric perspective (Fan et al., 2024; Wang et al., 2025a; Zhao et al., 2025), which holds significant potential for advancing the development of embodied intelli-

---

[*]Equal contribution   [1]Department of Electrical and Electronic Engineering, The Hong Kong Polytechnic University, Hong Kong SAR. [2]Division of Emerging Interdisciplinary Areas (EMIA), The Hong Kong University of Science and Technology, Hong Kong SAR. Correspondence to: Yi Wang[†] <yi-eie.wang@polyu.edu.hk>.

gent agents, such as assistive robotics and visual assistants.

Earlier approaches to egocentric hand-environment interaction understanding have predominantly addressed individual tasks in isolation. For example, action recognition (Peirone et al., 2025; Nasirimajd et al., 2025; Lyu et al., 2025) focuses on classifying action types, image captioning (Shi et al., 2025) aims to describe the interaction using sentences, and human-object interaction (HOI) detection (Su et al., 2025a; Chen et al., 2026; Deng et al., 2026; Su et al., 2026; Li et al., 2025a; Wang et al., 2025c) predicts bounding boxes or masks for human hands and the active objects involved. With the evolution of this field, researchers have increasingly shifted towards unified, user-centered tasks (Lai et al., 2024; Su et al., 2025c; Chen et al., 2023) to enable a deeper and more comprehensive understanding of interactive behavior from the first-person perspective. In this context, this paper concentrates on the task of egocentric interaction reasoning and grounding (Ego-IRG), which seeks to provide a structured analysis of interaction at three levels: (i) generating holistic textual descriptions of human-environment interactions in egocentric images; (ii) providing relevant textual responses to interaction-related queries posed by users; and (iii) producing precise, pixel-level grounding masks for the specific regions referenced in the queries.

As multimodal large language models (MLLMs) (Suglia et al., 2024; Kim et al., 2025; Shen et al., 2025) have demonstrated outstanding performance across various tasks, recent research (Su et al., 2025c) has explored their application to Ego-IRG to achieve a comprehensive understanding of egocentric interaction. Although these approaches have shown promising results, they still face notable limitations in interaction parsing accuracy, fine-grained grounding reliability, and cross-scene generalization capability. Meanwhile, recent studies (Wu et al., 2024; Mnih et al., 2024; Rafailov et al., 2023) have shown that reinforcement learning (RL) can significantly enhance the reasoning and generalization capabilities of MLLMs through reward-based feedback. Motivated by these, we propose EARL (**E**gocentric **A**nalysis-guided **RL**-based method), which integrates RL with MLLMs to jointly perform both textual reasoning and pixel-level grounding to understand egocentric interactions comprehensively.

EARL adopts a coarse-to-fine, two-stage framework comprising coarse-grained interpretation and fine-grained response. The former holistically interprets egocentric interactions and generates a structured textual description of them. The latter produces both a natural language answer and the corresponding pixel-level grounding mask in response to the user query. Rather than treating the two stages independently, we explicitly leverage the coarse analytical information to guide the query-oriented fine-grained response. Specifically, we extract a *global interaction descriptor* from the coarse-grained interpretation process and treat it as a semantic prior. This descriptor is then fused with multimodal features through a novel **A**nalysis-guided **F**eature **S**ynthesizer (**AFS**) to support subsequent fine-grained reasoning. Furthermore, to effectively train the network, EARL introduces a sophisticated, multi-faceted reward mechanism that incorporates format correctness, answer relevance, and grounding accuracy utilizing Group Relative Policy Optimization (GRPO) (Shao et al., 2024a). Our method achieves remarkable state-of-the-art (SOTA) performance on the Ego-IRGBench dataset (Su et al., 2025c) (Fig. 1), particularly in the grounding subtask, where it attains a cIoU of 65.48%, a 8.37% improvement over previous SOTA methods. Furthermore, out-of-distribution (OOD) grounding experiments on the EgoHOS (Zhang et al., 2022a) dataset further highlight the outstanding generalization capability of our method. To summarize, our work has the main contributions:

- We propose EARL, a two-stage subtask-collaborative framework for Ego-IRG, which explicitly transfers coarse interaction semantics from holistic analysis to fine-grained answering and grounding rather than treating the subtasks as independent parallel objectives.

- We introduce an Analysis-guided Feature Synthesizer (AFS) to address the noisy-prior problem induced by the first-stage analysis. Instead of indiscriminately fusing analysis features, AFS follows a selection-and-fusion paradigm that first refines the global interaction descriptor and then injects reliable semantic priors into query-conditioned response generation.

- We design a multi-faceted GRPO-based optimization strategy for heterogeneous Ego-IRG outputs, jointly considering format correctness, answer relevance, and grounding accuracy to align textual reasoning with pixel-level localization.

- Extensive experiments demonstrate the effectiveness of our method, achieving substantial in-domain improvements of 1.682 CIDEr and 8.37% cIoU in answering and grounding subtasks, respectively. Furthermore, OOD evaluations on the EgoHOS dataset further prove the generalization capability of our approach.

## 2. Related Work

### 2.1. Egocentric Interaction Understanding

Egocentric interaction understanding has emerged as a pivotal research area in computer vision, focusing on how individuals interact with their environment from a first-person perspective. Early studies in this field primarily focused on isolated tasks, such as action recognition and object detection (Bambach et al., 2015; Zhang et al., 2022b; Leonardi et al., 2022). For instance, Wang et al. (Wang et al., 2023;

Li et al., 2022) introduced methods for classifying verbs and nouns in egocentric videos and images, enabling models to achieve a coarse understanding of human interactions. To improve recognition performance, subsequent research has emphasized the incorporation of action-related cues such as hands, active objects, and their spatial relationships. For example, Kwon et al. (Kwon et al., 2021) introduced the H2O dataset, which focuses on predicting hand and object poses to enhance the recognition of egocentric interactions. Moreover, Wang et al. (Wang et al., 2020a;b; Shan et al., 2020; Lu & Mayol-Cuevas, 2021; Shiota et al., 2024; Yu et al., 2023; Li et al., 2019; Liu et al., 2021; Wang et al., 2020a) demonstrated that integrating active object detection can lead to a more accurate understanding of ongoing interactions. These findings underscore the importance of analyzing multifaceted cues for the precise understanding of egocentric human-environment interactions. In addition to visual features, researchers have explored integrating multiple modalities to enhance recognition capabilities. For instance, Liu et al. (Liu et al., 2020; Sudhakaran et al., 2019; Shen et al., 2018; Zatsarynna et al., 2021; Wang et al., 2021) investigated the use of gaze and audio signals to further improve interaction understanding. However, these studies lack a comprehensive understanding of interactions, such as generating fluent textual descriptions and fine-grained mask responses. Additionally, their results cannot be directly applied to multiple downstream tasks with varying requirements or queries. Unlike the aforementioned studies, this paper focuses on the Ego-IRG task (Su et al., 2025c), which aims to comprehensively parse the egocentric interaction from three aspects, i.e., generating the analytical descriptions, textual answers, and grounding mask responses based on user queries, which enables a deeper understanding of egocentric interactions and can be readily applied to downstream tasks with diverse user queries.

### 2.2. Reinforcement Learning for MLLMs

Reinforcement learning (Sutton & Barto, 1998) has proven effective for improving the training of large multimodal models (LMMs). Among the various RL techniques (Rowland et al., 2024; Rafailov et al., 2023; Mnih et al., 2015), Group Relative Policy Optimization (Shao et al., 2024b) stands out for its efficiency and potential. Unlike conventional RL methods that rely on a critic model to estimate the baseline, GRPO derives the baseline from group scores, significantly reducing the computational resources required during training. This lower resource demand makes GRPO an appealing choice for training large-scale models. Recent studies (Shen et al., 2025; Wang et al., 2025b) have demonstrated that integrating GRPO into LMMs yields notable improvements in visual generation and understanding tasks. For example, Seg-Zero (Liu et al., 2025b) combines GRPO with MLLMs for segmentation reasoning, enabling

explicit chain-of-thought (CoT) reasoning through cognitive reinforcement. Similarly, Seg-R1 (You & Wu, 2025) incorporates GRPO into the segmentation domain, equipping LMMs with pixel-level understanding via a carefully designed training strategy. SAM-R1 (Huang et al., 2025) further advances fine-grained reasoning in image understanding tasks by leveraging task-specific, detailed rewards and a customized optimization objective, using the Segment Anything Model (SAM) (Ravi et al., 2025) as the reward provider. In this study, we extend the application of GRPO to egocentric interaction reasoning and grounding tasks, providing a more effective approach for enhancing MLLMs' performance in comprehending egocentric interactions.

## 3. Methodology

### 3.1. Task Definition

Taking an egocentric image $\mathcal{I}$ and a user query $T_q$ as input, this paper focus on unified egocentric interaction reasoning and grounding (Ego-IRG) task, which aims to: (i) generating a holistic textual description $T_{ana}$ of human-environment interactions in egocentric images; (ii) providing textual responses $T_{ans}$ containing the entities that need to be segmented in response to the query; and (iii) producing precise pixel-level grounding masks $M$ for the targeted regions referenced in the queries.

### 3.2. Overview of EARL

This work presents a two-stage framework, EARL, designed for unified parsing of egocentric interactions ranging from image-level descriptions to instance-level grounding, as illustrated in Fig. 2.

**Coarse-grained Interpretation.** The first stage, i.e., coarse-grained interpretation, is dedicated to achieving a thorough and integrated understanding of human–object interactions in egocentric images, with the goal of producing a textual analysis that characterizes the ongoing interaction. The inputs to this stage consist of an egocentric image $\mathcal{I}$ and a general analysis instruction $T_a$. In our implementation, $T_a$ is instantiated as: "Please analyze the interactions of hands and objects in detail". Given these inputs, the model first extracts rich multimodal representations using a visual encoder $\mathcal{E}_v$ and a text encoder $\mathcal{E}_t$. These embeddings are then processed by an analyzing VLM decoder $\mathcal{D}_{vlm}$, which outputs a precise and contextually informed textual description $T_{ana}$ that captures the contextual relationships among hands, objects, and actions. The coarse-grained interpretation is supervised by $\mathcal{L}_{des}$, computed as the cross-entropy loss between the predicted analysis and the ground truth. Furthermore, since the generated analysis encapsulates rich semantic information from the image, we leverage its corresponding feature representation to enhance perfor-

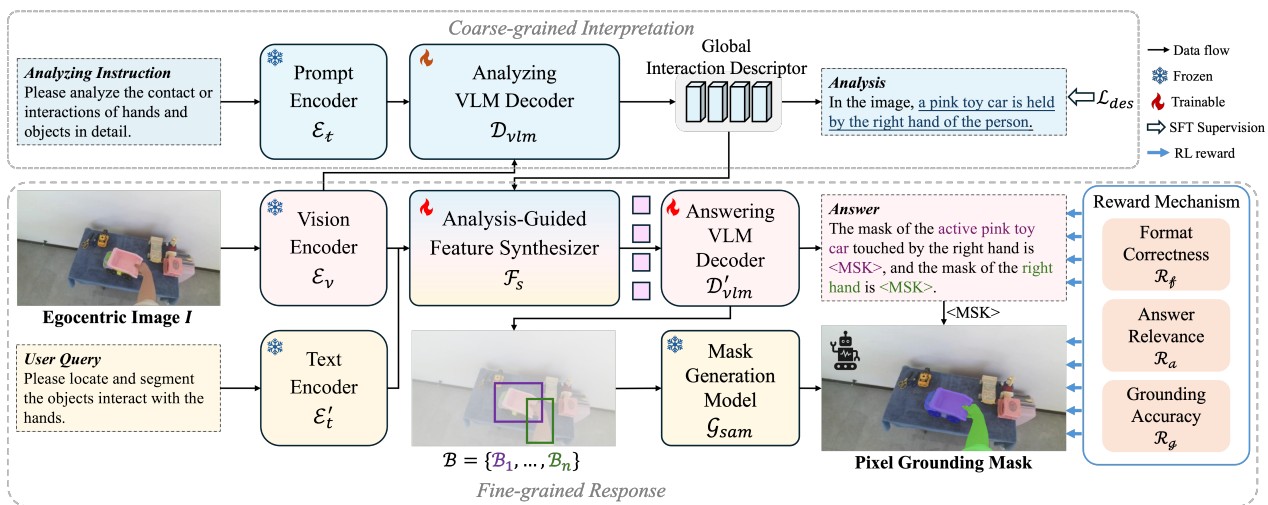

*Figure 2.* **The architecture of our proposed EARL.** Our approach combines the analysis-guided reinforcement learning with MLLMs to achieve a comprehensive interpretation of human-environment interactions from an egocentric perspective.

mance in the subsequent subtasks. Specifically, we retain the last-layer hidden embeddings from $\mathcal{D}_{vlm}$ as a *global interaction descriptor*, denoted as $\mathbf{F}_{ana}$, which serves as prior knowledge to inform both answer generation and pixel-level grounding. This design ensures that high-level contextual cues guide subsequent instance-level downstream task execution. The process of the coarse-grained interpretation can be denoted as follows:

$$T_{ana}, \mathbf{F}_{ana} = \mathcal{D}_{vlm}\left(\mathcal{E}_v(\mathcal{I}), \mathcal{E}_t(T_a)\right). \quad (1)$$

We first train the coarse-grained interpretation model to generate an accurate and holistic textual description $T_{ana}$ of human–environment interactions in egocentric images. As a result, the corresponding global interaction descriptor $\mathbf{F}_{ana}$, which encodes rich contextual semantics, becomes sufficiently reliable to serve as a semantic prior for supporting subsequent query-oriented subtasks.

**Fine-grained Response.** The fine-grained response module is designed to produce both a textual answer $T_{ans}$ and pixel-level grounding masks $\mathcal{M}$ in response to a user query, guided by $\mathbf{F}_{ana}$. Specifically, given an egocentric image $\mathcal{I}$ and a user query $T_q$, the model first encodes the image and the query using a vision encoder $\mathcal{E}_v$ and a text encoder $\mathcal{E}'_t$, respectively. To fully leverage semantic cues from the analysis, we propose an **analysis-guided feature synthesizer (AFS)** that fuses encoded features with $\mathbf{F}_{ana}$ to produce a unified multimodal representation $\mathbf{F}_R$. This integration enables the effective combination of visual, textual, and high-level analytical information, thereby facilitating the generation of accurate, context-aware responses. The detailed architecture and mechanism of the AFS are elaborated in Section 3.3. The overall feature extraction process can be formulated as:

$$\mathbf{F}_R = \mathcal{F}_s\left(\mathcal{E}_v(\mathcal{I}), \mathcal{E}'_t(T_q), \mathbf{F}_{ana}\right), \quad (2)$$

where the $\mathcal{F}_s$ denotes the AFS module.

Subsequently, we utilize an answering VLM decoder $\mathcal{D}'_{vlm}$ to generate a textual response $T_{ans}$ conditioned on the unified context-aware multimodal representation $\mathbf{F}_R$. In addition to the textual answer, we prompt $\mathcal{D}'_{vlm}$ to predict bounding boxes $\mathcal{B} = \{\mathcal{B}_1, \ldots, \mathcal{B}_n\}$, where each $\mathcal{B}_i = (s^i_x, s^i_y, e^i_x, e^i_y)$ corresponds to an entity mentioned in $T_{ans}$. Here, $n$ denotes the number of referred entities, and $s^i_x, s^i_y, e^i_x, e^i_y$ represent the coordinates of the $i-th$ bounding box. These predicted bounding boxes are then passed to a mask generation model $\mathcal{G}_{sam}$, which outputs the corresponding pixel-level grounding masks $\mathcal{M}$. In practice, we employ Qwen2.5VL-7B (Bai et al., 2025) as the answering VLM decoder, and adopt SAM2 (Ravi et al., 2024) for mask generation. The overall procedure of the fine-grained response module can be summarized as follows:

$$T_{ans}, \mathcal{B} = \mathcal{D}'_{vlm}(\mathbf{F}_R), \quad \mathcal{M} = \mathcal{G}_{sam}(\mathcal{B}, \mathcal{I}). \quad (3)$$

Motivated by evidence that RL can significantly improve the reasoning and generalization performance of MLLMs through reward-driven feedback, we introduce an RL-based training strategy for the second-stage model. Specifically, we employ Group Relative Policy Optimization (GRPO) to optimize the network during this stage. In RL, reward functions play a crucial role in defining the optimization objective and steering the model toward desired behaviors and outcomes. To encourage outputs that are structurally valid, semantically accurate, and precisely grounded, we design a reward function consisting of three components:

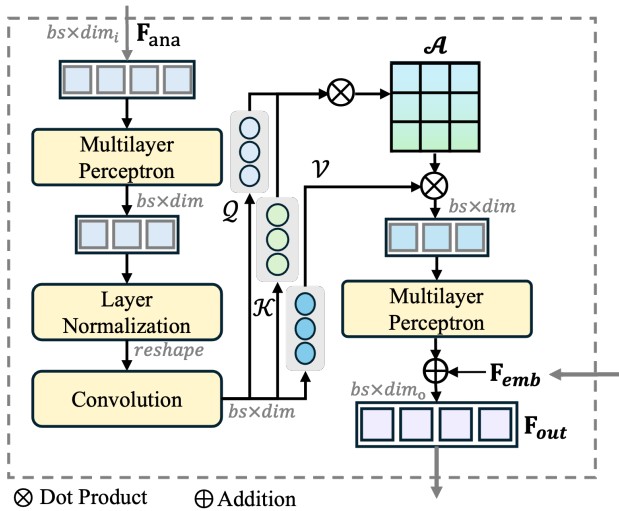

*Figure 3.* **Detailed architecture of the proposed AFS.**

the format correctness reward $\mathcal{R}_f$, the answer relevance reward $\mathcal{R}_a$, and the grounding accuracy reward $\mathcal{R}_g$. The formulation and implementation details of these rewards are provided in Section 3.4.

### 3.3. Analysis-guided Feature Synthesizer

As described in Sec. 3.2, we propose an analysis-guided feature synthesizer (AFS) to address a key challenge in our two-stage framework: how to effectively bridge coarse-grained interaction interpretation with fine-grained query-oriented reasoning.

To address this problem, we propose an analysis-guided feature synthesizer to integrate global interaction descriptor $\mathbf{F}_{ana} \in \mathbb{R}^{bs \times dim_i}$ with encoded visual features $\mathbf{F}_v$ and textual features $\mathbf{F}_t$, aiming to construct a unified multimodal representation $\mathbf{F}_R$. Here, $bs$ denotes the batch size, and $dim_i$ represents the feature dimension of $\mathbf{F}_{ana}$. The detailed architecture of the feature synthesizer is illustrated in Fig. 3.

Prior to the feature synthesizer, we utilize Qwen2.5-VL (Bai et al., 2025) to align and fuse the encoded visual features $\mathbf{F}_v$ and textual features $\mathbf{F}_t$, resulting in $\mathbf{F}_{emb} \in \mathbb{R}^{bs \times dim_o}$, where the $dim_o$ is the feature dimension of $\mathbf{F}_{emb}$. Subsequently, our feature synthesizer employs a multilayer perceptron (MLP) $\phi_m(\cdot)$ to reduce the dimensionality of the analysis feature $\mathbf{F}_{ana}$ to $dim$, followed by layer normalization $\phi_{ln}(\cdot)$. Next, $\mathbf{F}_{ana}$ is reshaped into $bs \times h \times w$, where $h = w = \sqrt{dim}$, and the reshaped feature is passed through a convolutional layer $\phi_c(\cdot)$ to generate the query $\mathcal{Q}$, key $\mathcal{K}$, and value $\mathcal{V}$. Moreover, the $\mathcal{Q}, \mathcal{K}$, and $\mathcal{V}$ are then reshaped back to $bs \times dim$. The analysis-guided self-attention operation is subsequently performed, projecting holistic scene understanding into the response space and forming a cohe-

sive foundation for answer generation and visual grounding. The process of the analysis-guided self-attention can be described as:

$$\mathcal{Q}, \mathcal{K}, \mathcal{V} = \phi_c(\phi_{ln}(\phi_m(\mathbf{F}_{ana}))), \quad (4)$$

$$\mathbf{F} = \text{softmax}\left(\frac{\mathcal{Q}\mathcal{K}^\top}{\sqrt{dim}}\right)\mathcal{V}, \quad (5)$$

where $\mathbf{F}$ is the output feature of the self-attention operation with the dimension of $bs \times dim$. Then, another MLP $\phi'_m(\cdot)$ is used to project the dimension of $\mathbf{F}$ to match $\mathbf{F}_{emb}$. Finally, the summation of $\mathbf{F}$ and $\mathbf{F}_{emb}$ is performed to obtain the fused feature $\mathbf{F}_{out}$, which serves as the input to the answering VLM decoder $\mathcal{D}'_{vlm}$, i.e., $\mathbf{F}_{out} = \mathbf{F}_{emb} + \phi'_m(\mathbf{F})$. The resulting representation $\mathbf{F}_{out}$ is then fed into the answer-generating VLM decoder $\mathcal{D}'_{vlm}$ to predict both the textual answer and corresponding bounding boxes.

Through this simple yet effective AFS, the global interaction descriptor serves as a semantic prior for subsequent subtasks, delivering both rich and precise contextual information. It thus bridges multiple subtasks while enhancing the contextual coherence of predictions.

### 3.4. Reward Design

The fine-grained response module is optimized using Group Relative Policy Optimization (GRPO). During RL training, the vision encoder $\mathcal{E}_v$, text encoder $\mathcal{E}'_t$, and mask generation model $\mathcal{G}_{sam}$ remain frozen. Reward design plays a central role in reinforcement learning, as it defines the optimization objective and guides the model toward desired behaviors. In this section, we describe the three-component reward function used in our framework: a format correctness reward $\mathcal{R}_f$, an answer relevance reward $\mathcal{R}_a$, and a grounding accuracy reward $\mathcal{R}_g$.

**Format Correctness Reward.** The format correctness reward encourages the model to adhere to a predefined output structure that includes both `<answer>` and `<bbox>` tags, which ensures that the generated responses are not only informative but also easily parsable for downstream tasks. It is defined as:

$$R_f = \begin{cases} 1, & \text{valid structure} \\ 0.5, & \text{partly matched} \\ 0, & \text{otherwise.} \end{cases} \quad (6)$$

A valid structure follows the expected XMI-like format, facilitating automated parsing and evaluation. By enforcing this structural consistency, the format correctness reward helps maintain the integrity of the model's outputs, ensuring that they can be reliably interpreted and utilized in subsequent processing stages.

**Answer Relevance Reward.** The answer relevance reward evaluates the textual consistency between the predicted an-

*Table 1.* **Comparison results on Ego-IRGBench test and val sets.** Text highlighted in blue indicates the difference between our method and the **best-performing method**, while text in red represents the improvement of our approach over the second-best method.

| Model | Type | Test | | | | | Val | | | | |
|---|---|---|---|---|---|---|---|---|---|---|---|
| | | Analysis | | Answering | | Grounding | Analysis | | Answering | | Grounding |
| | | M↑ | C↑ | M↑ | C↑ | cIoU↑ | M↑ | C↑ | M↑ | C↑ | cIoU↑ |
| *General-domain MLLMs* | | | | | | | | | | | |
| Qwen3-VL-4B (Yang et al., 2025) | Gen | 0.134 | 0.041 | 0.882 | 4.945 | 22.84 | 0.136 | 0.045 | 0.882 | 5.001 | 23.31 |
| Qwen3-VL-8B (Yang et al., 2025) | Gen | 0.136 | 0.044 | 0.882 | 5.000 | 23.31 | 0.188 | 0.119 | 0.898 | 4.540 | 24.05 |
| Qwen2.5-VL-3B (Bai et al., 2025) | Gen | 0.158 | 0.071 | 0.627 | 2.423 | 12.44 | 0.158 | 0.071 | 0.627 | 2.416 | 12.51 |
| Qwen2.5VL-7B (Bai et al., 2025) | Gen | 0.206 | 0.119 | 0.739 | 2.477 | 23.71 | 0.204 | 0.119 | 0.729 | 2.417 | 22.46 |
| InternVL2.5-7B (Chen et al., 2024) | Gen | 0.177 | 0.044 | 0.681 | 1.533 | 27.21 | 0.176 | 0.047 | 0.679 | 1.547 | 27.70 |
| *Egocentric MLLMs* | | | | | | | | | | | |
| EgoThinker (Pei et al., 2025) | Ego | 0.182 | 0.061 | 0.491 | 2.344 | 12.33 | 0.184 | 0.065 | 0.495 | 2.364 | 12.87 |
| ANNEXE* (Su et al., 2025b) | Ego | 0.536 | 1.423 | 0.350 | 2.317 | 31.90 | 0.537 | 1.447 | 0.352 | 2.312 | 31.72 |
| ANNEXE (Su et al., 2025b) | Ego | **0.563** | 1.494 | 0.365 | 2.590 | 36.02 | **0.563** | 1.516 | 0.363 | 2.543 | 35.14 |
| *Pixel-level Grounding MLLMs* | | | | | | | | | | | |
| Sa2VA-InternVL3-8B (Yuan et al., 2025) | Gen | 0.207 | 0.192 | 0.486 | 2.309 | 31.17 | 0.280 | 0.198 | 0.489 | 2.320 | 37.97 |
| Sa2VA-8B (Yuan et al., 2025) | Gen | 0.188 | 0.115 | 0.754 | 2.656 | 32.69 | 0.189 | 0.120 | 0.448 | 2.644 | 34.89 |
| UniPixel-7B (Liu et al., 2025a) | Gen | 0.165 | 0.075 | 0.158 | 1.136 | 14.01 | 0.168 | 0.065 | 0.155 | 1.128 | 13.63 |
| *Reinforcement Learning-based General Segmentation MLLMs* | | | | | | | | | | | |
| Seg-R1-3B (You & Wu, 2025) | Gen | 0.268 | 0.368 | 0.696 | 1.914 | 28.91 | 0.266 | 0.367 | 0.695 | 1.929 | 29.08 |
| Seg-R1-7B (You & Wu, 2025) | Gen | 0.292 | 0.289 | 0.774 | 2.483 | 46.10 | 0.289 | 0.285 | 0.771 | 2.488 | 45.05 |
| Seg-Zero (Liu et al., 2025b) | Gen | 0.309 | 0.049 | 0.742 | 2.380 | 57.11 | 0.310 | 0.052 | 0.740 | 2.349 | 58.03 |
| **EARL (Ours)** | Ego | 0.542 | **1.522** | **0.939** | **6.682** | **65.48** | 0.541 | **1.520** | **0.933** | **6.610** | **62.71** |
| | | −0.021 | +0.028 | +0.057 | +1.682 | +8.37 | −0.022 | +0.004 | +0.035 | +1.609 | +4.68 |

swer $T_{ans}$ and the ground-truth answer $T_{ans}^{gt}$. It combines exact match and semantic similarity to balance lexical precision and expressive flexibility:

$$R_a = \text{ExactMatch}(T_{ans}, T_{ans}^{gt}) + \text{SemanticSim}(T_{ans}, T_{ans}^{gt}), \quad (7)$$

where $\text{ExactMatch}(\cdot)$ returns 1 if the predicted answer is identical to the ground-truth answer, and 0 otherwise. Following previous work (Shen et al., 2025), $\text{SemanticSim}(\cdot)$ is computed using the Levenshtein ratio (Bard, 2007), measuring how closely the predicted answer aligns with the reference in meaning, even if the wording differs.

**Grounding Accuracy Reward.** The grounding accuracy reward quantifies the spatial alignment between the predicted region and the ground-truth mask $M_{gt}$. It can be computed as:

$$R_g = \text{IoU}(\mathcal{M}, M_{gt}), \quad (8)$$

where IoU (Intersection over Union) (Su et al., 2021) measures the degree of overlap between the predicted mask $\mathcal{M}$ and the ground-truth mask $M_{gt}$. A higher IoU value indicates a greater correspondence between the predicted and actual regions. We introduce three weighting factors, $\lambda_f$, $\lambda_a$, and $\lambda_g$, corresponding to the format correctness reward $R_f$, the answer relevance reward $R_a$, and the grounding accuracy reward $R_g$, respectively. The final reward function used to train the response module is formulated as a weighted sum of these individual reward components. By incorporating this reward, the model is able to generate predictions that are not only semantically relevant but also precisely grounded at the pixel level, thereby enhancing its ability to perform fine-grained spatial reasoning and object localization. More details of the training strategy can be found in Sec. A.2.

## 4. Experiments

We evaluate the proposed EARL framework on both in-domain and out-of-distribution benchmarks for egocentric interaction understanding. We first introduce the datasets, evaluation metrics, and implementation details used in our experiments. We then present quantitative comparisons, ablation and additional analyses, followed by qualitative visualizations on the Ego-IRGBench dataset.

### 4.1. Datasets, Metrics, and Implementation Details

**Training and In-Domain Testing Dataset.** We use the Ego-IRGBench (Su et al., 2025b) dataset to train our model and evaluate its in-domain performance. Ego-IRGBench comprises 20,504 egocentric images collected from the HOI4D (Liu et al., 2022) dataset. Each egocentric RGB image is paired with a depth map and an interaction description specifying the ongoing interaction. In addition, multiple interaction-related queries are annotated for each image, and each query is paired with an answer and a corresponding pixel-level grounding mask.

**Out-of-Distribution Testing Dataset.** Given the limited availability of unified egocentric datasets for all subtasks, we use EgoHOS (Zhang et al., 2022a) as the out-of-distribution (OOD) evaluation set. EgoHOS contains 11,743 egocentric images with per-pixel segmentation labels for hands and interacting objects. Since the annotations in EgoHOS only support the grounding subtask, our OOD evaluation focuses exclusively on pixel-level grounding, which is also the ultimate objective of the Ego-IRG task. More details about the datasets are provided in Appendix A.1.

*Table 2.* **Out-of-distribution grounding comparison on the EgoHOS dataset.** Text in **bold** and underlined indicates the best and second-best results, respectively. The value in red denotes the improvement of our method over the second-best method in overall cIoU.

| Method | Type | Left Hand IoU (%) ↑ | Right Hand IoU (%) ↑ | Left-hand Objects IoU (%) ↑ | Right-hand Objects IoU (%) ↑ | Two-hand Objects IoU (%) ↑ | cIoU (%) ↑ |
|---|---|---|---|---|---|---|---|
| *Referring Image Segmentation Methods* | | | | | | | |
| X-Decoder (2023) (Zou et al., 2023) | Gen | 18.19 | 20.68 | 5.37 | 7.99 | 11.21 | 12.69 |
| GSVA (2024) (Xia et al., 2024) | Gen | 20.69 | 23.18 | 5.37 | 7.79 | 9.02 | 13.21 |
| GRES (2023) (Liu et al., 2023) | Gen | 20.49 | 22.98 | 7.22 | 8.64 | 10.87 | 14.04 |
| LISA (2024) (Lai et al., 2024) | Gen | 28.93 | 33.06 | 14.27 | 17.94 | 18.10 | 22.46 |
| *General-domain MLLMs* | | | | | | | |
| InternVL3-8B (Chen et al., 2024) | Gen | 12.78 | 9.97 | 4.65 | 0.06 | 3.09 | 6.11 |
| Qwen2.5-VL-3B (Bai et al., 2025) | Gen | 5.95 | 23.54 | 0.18 | 0.54 | 0.47 | 6.14 |
| Qwen3-VL-4B (Yang et al., 2025) | Gen | 15.68 | 9.16 | 5.79 | 4.24 | 6.83 | 8.34 |
| Qwen2.5-VL-7B (Bai et al., 2025) | Gen | 23.20 | 16.45 | 4.74 | 5.82 | 12.58 | 12.56 |
| Qwen3-VL-8B (Yang et al., 2025) | Gen | 30.70 | 27.75 | 14.16 | 20.53 | 26.61 | 23.95 |
| *Egocentric MLLMs* | | | | | | | |
| EgoThinker (Pei et al., 2025) | Ego | 29.33 | 25.09 | 14.01 | 15.81 | 22.84 | 21.42 |
| ANNEXE (Su et al., 2025b) | Ego | 25.84 | 26.21 | 16.24 | 15.86 | 22.75 | 21.38 |
| *Pixel-level Grounding MLLMs* | | | | | | | |
| Sa2VA-8B (Yuan et al., 2025) | Pix | 48.56 | 45.82 | 26.91 | 26.77 | 37.04 | 37.63 |
| Sa2VA-InternVL3-8B (Yuan et al., 2025) | Pix | 41.24 | 41.47 | 25.95 | 26.54 | **37.45** | 34.08 |
| UniPixel-7B (Liu et al., 2025a) | Pix | 28.54 | 24.82 | 11.12 | 12.45 | 19.06 | 21.14 |
| *Reinforcement Learning-based General Segmentation MLLMs* | | | | | | | |
| Seg-Zero (Liu et al., 2025b) | Gen | 33.28 | 30.56 | 15.86 | 14.75 | 33.28 | 25.55 |
| Seg-R1-3B (You & Wu, 2025) | Gen | 37.72 | 30.76 | 21.17 | 21.97 | 27.23 | 27.77 |
| Seg-R1-7B (You & Wu, 2025) | Gen | 41.47 | 40.08 | **29.86** | **28.40** | 34.94 | 34.95 |
| **EARL (Ours)** | Ego | **52.30** | **62.44** | 19.10 | 20.57 | 22.85 | **38.21**₊₀.₅₈ |

**Metrics.** During evaluation, we assess three aspects: (i) the quality of generated interaction descriptions, (ii) the quality of generated answers to queries, and (iii) pixel-level grounding accuracy. Following previous work (Su et al., 2025b), we adopt METEOR (Banerjee & Lavie, 2005) and CIDEr (Vedantam et al., 2015) to evaluate interaction descriptions and answers. For pixel grounding, we use cIoU (Su et al., 2025c).

**Implementation Details.** Unless otherwise stated, we use Qwen2.5-VL-7B (Bai et al., 2025) as the answering VLM decoder and Qwen2.5-VL-3B (Bai et al., 2025) as the analyzing VLM decoder. Our implementation consists of two sequentially trained stages. The vision encoders are frozen in both stages. Stage 1 is trained with supervised fine-tuning to generate holistic interaction descriptions and reliable global interaction descriptors. Stage 2 is trained with Stabilized GRPO to generate query-conditioned textual answers and bounding boxes, while SAM2-Large is kept frozen and used only to convert predicted boxes into pixel-level masks. More implementation details are provided in Appendix A.3.

### 4.2. Quantitative Results

**In-Domain Comparison Results.** To evaluate the effectiveness of EARL for comprehen- sive egocentric interaction understanding, we compare it against several categories of baselines on the Ego-IRGBench dataset, including general-domain MLLMs (Yang et al., 2025; Bai et al., 2025; Chen et al., 2024), egocentric MLLMs (Pei et al., 2025; Su et al.,

*Table 3.* **Ablation study on the Ego-IRGBench test set.** We compare different training strategies and fusion mechanisms. "–" denotes that no additional analysis-to-task fusion is used, and CAF denotes cross-attention fusion.

| Method | | Analysis | | Answering | | Grounding |
|---|---|---|---|---|---|---|
| Train | Fusion | M | C | M | C | cIoU |
| SFT | – | 0.538 | 1.421 | 0.362 | 2.339 | 32.47 |
| SFT | AFS | 0.540 | 1.457 | 0.498 | 4.313 | 43.85 |
| RL | – | 0.370 | 0.927 | 0.441 | 2.703 | 39.81 |
| RL | Concat | 0.398 | 1.132 | 0.443 | 2.715 | 39.88 |
| RL | Sum | 0.449 | 1.378 | 0.674 | 2.807 | 37.14 |
| RL | MLP | 0.371 | 1.290 | 0.412 | 2.701 | 37.29 |
| RL | CAF | 0.502 | 1.487 | 0.790 | 5.241 | 52.36 |
| RL | AFS | **0.542** | **1.522** | **0.939** | **6.682** | **65.48** |

2025b), pixel-level grounding MLLMs (Yuan et al., 2025; Liu et al., 2025a), and reinforcement learning-based general segmentation MLLMs (You & Wu, 2025; Liu et al., 2025b). The quanti- tative results are summarized in Table 1.

As shown, EARL achieves the best performance on nearly all metrics across both the test and validation sets. On the test set, although its analysis METEOR score (0.542) is slightly lower than that of ANNEXE (0.563), EARL attains the best analysis CIDEr (1.522), answering ME-TEOR/CIDEr (0.939/6.682), and grounding cIoU (65.48). In particular, the grounding performance surpasses the second-best baseline by 8.37 points, demonstrating a strong ability to identify and localize interaction-relevant entities from queries. Similar trends are observed on the validation set, where EARL consistently delivers the strongest performance in answering and grounding.

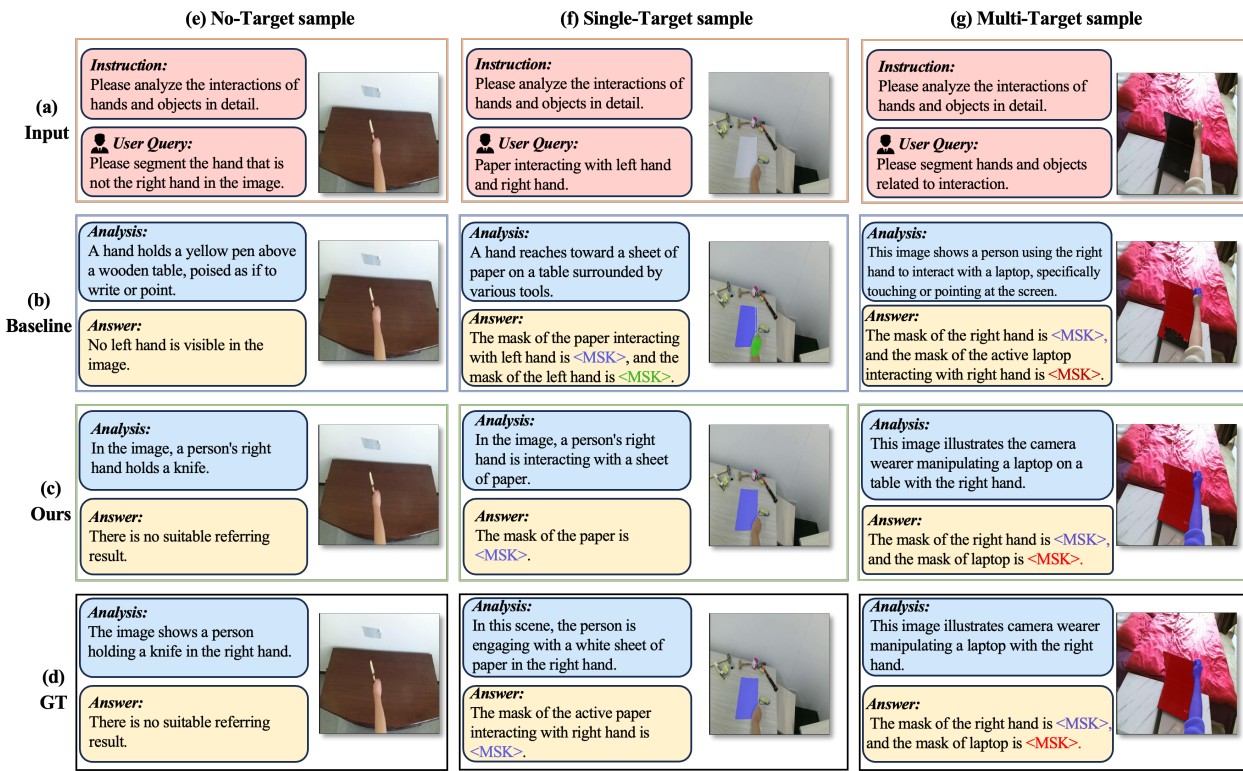

*Figure 4.* **Visualization results of our methods on Ego-IRGBench dataset.**

*Table 4.* **Hyperparameter study results on Ego-IRGBench test set.** Text in **bold** indicates the best-performing method.

| # | $\lambda_f$ | $\lambda_a$ | $\lambda_g$ | Ana | | Ans | | Gro |
|---|---|---|---|---|---|---|---|---|
| | | | | M | C | M | C | cIoU |
| 1 | 1 | 1 | 0 | 0.539 | 1.518 | 0.926 | 6.380 | 37.99 |
| 2 | 1 | 0 | 1 | 0.541 | 1.520 | 0.852 | 4.862 | 58.22 |
| 3 | 0.5 | 1 | 1 | 0.539 | 1.520 | 0.921 | 6.356 | **69.87** |
| 4 | 1 | 1 | 1 | **0.542** | **1.522** | **0.939** | **6.682** | 65.48 |

These gains can be attributed to two complementary designs in EARL. First, reinforcement learning improves reasoning and grounding through structured reward signals. Second, the proposed AFS transfers semantic priors extracted from the analysis stage to the answering and grounding stages, effectively bridging high-level interaction understanding and fine-grained visual localization. Together, these components enable EARL to achieve a more holistic and precise understanding of egocentric interactions.

**OOD Comparison Results on Grounding.** To evaluate the generalization ability of our method, we conduct out-of-distribution (OOD) evaluation on the EgoHOS dataset. Since EgoHOS does not provide textual annotations compatible with Ego-IRG, we focus exclusively on the grounding subtask, which is also a core objective of our framework. We evaluate grounding performance on the entire EgoHOS

dataset without distinguishing between its original training and test splits.

For comparison, we include several groups of baselines, including referring image segmentation methods (Xia et al., 2024; Lai et al., 2024; Liu et al., 2023; Zou et al., 2023), general-domain MLLMs (Yang et al., 2025; Bai et al., 2025; Chen et al., 2024), egocentric MLLMs (Pei et al., 2025; Su et al., 2025b), pixel-level grounding MLLMs (Yuan et al., 2025; Liu et al., 2025a), and reinforcement learning-based general segmentation MLLMs (You & Wu, 2025; Liu et al., 2025b). The quantitative results are reported in Table 2.

As shown, our method achieves the best overall cIoU of 38.21%, outperforming the second-best method by 0.58 points. In particular, our approach yields clear advantages in hand grounding, reaching 52.30% IoU for the left hand and 62.44% IoU for the right hand. Although some specialized grounding or RL-based segmentation models perform better on certain object-centric categories, our method delivers the strongest overall OOD grounding performance. These results suggest that the proposed reward design improves robustness under distribution shift and enhances the model's ability to generalize to unseen egocentric interaction scenarios.

**Ablation Study.** As described in Sec. 3.2, we incorporate reinforcement learning to strengthen the reasoning and

grounding capabilities of MLLMs, and introduce an AFS to project analysis embeddings into the answer and grounding spaces. In this section, we present ablation studies on both the RL component and the AFS to demonstrate their effectiveness. The results are summarized in Table 3. In the reported results, the baseline uses Qwen2.5-VL (Bai et al., 2025) to generate analysis and answer outputs, while SAM2 (Ravi et al., 2024) produces the grounding masks. The results demonstrate that RL significantly enhances the grounding capabilities of MLLMs. Beyond assessing the impact of RL, we explore various strategies for fusing the global interaction descriptor with image and text features (Fusion), including MLP, cross-attention fusion (CAF), et al. Among the approaches evaluated, our proposed AFS consistently delivers superior performance. By integrating analysis knowledge, AFS empowers the model to generate more contextually relevant answers and produce more precise grounding masks, thereby enhancing overall task performance.

**Hyperparameter Study.** As described in Sec. 3.4, the weights assigned to the different reward functions, i.e., $\lambda_f$, $\lambda_a$, and $\lambda_g$, must be carefully set during training to balance the contributions of each component. In this section, we systematically investigate the effects of varying these hyperparameters on model performance, as shown in Table 4. In particular, we observe that removing either the answer similarity or grounding accuracy terms from the reward mechanism results in a pronounced decline in overall performance, which is straightforward to understand. Furthermore, when the weight on format correctness $\lambda_f$ is reduced to 0.5, there is a noticeable decrease in the performance on the answering and analysis tasks. However, this adjustment yields a significant improvement of 4.39% in grounding cIoU, indicating that prioritizing grounding accuracy can enhance localization capabilities, albeit at the expense of other aspects. These findings underscore the importance of carefully tuning the reward weights to achieve a balanced and robust model. The grounding accuracy and answer relevance rewards, in particular, are crucial for understanding egocentric interactions.

### 4.3. Qualitative Evaluation

To visualize the performance of our model, we compare our method with the baseline (Qwen2.5-VL plus SAM2) in Fig. 4. The comparison includes three representative sample types: no-target, single-target, and multi-target queries. As shown, our model produces analytical descriptions and answers that align more closely with the ground truth (GT). In multi-target scenarios, our method also accurately localizes the boundaries of each referred object, further validating its capability in fine-grained visual grounding. More visualizations are provided in Appendix A.5.

## 5. Discussion

### 5.1. Conclusion

In this work, we address the challenging task of egocentric interaction understanding, which requires unified image-level analysis, instance-level answering, and pixel-level grounding in response to user queries. To enhance the generalization ability of MLLMs for comprehensive interaction understanding, we propose EARL that integrates GRPO to improve both reasoning and visual grounding. We further introduce an AFS to transfer global semantic descriptors into fine-grained query-driven reasoning. A multi-faceted reward mechanism is designed to guide policy optimization, incorporating format correctness, answer relevance, and grounding accuracy. Extensive experiments on the Ego-IRGBench benchmark and out-of-distribution evaluations demonstrate the superior performance and strong generalization capability of EARL.

### 5.2. Limitations and Future Work

Despite its strong performance on Ego-IRGBench, EARL has several limitations. First, small or heavily occluded interactive objects may lead to inaccurate bounding box predictions and grounding masks. Second, our out-of-distribution (OOD) evaluation is currently limited to grounding, as existing egocentric OOD datasets (e.g., EgoHOS) do not provide unified annotations for analysis, answering, and grounding. Therefore, the reported OOD results should be interpreted as indicative of grounding transferability rather than full-task generalization. Third, EARL is specifically designed for egocentric interaction scenarios and may not directly generalize to third-person (exocentric) interaction benchmarks. Future work will explore multi-scale grounding refinement, the development of unified full-task OOD benchmarks, and architectures that support both egocentric and exocentric interaction reasoning.

## Acknowledge

The research work described in this paper was conducted in the JC STEM Lab of Machine Learning and Computer Vision funded by The Hong Kong Jockey Club Charities Trust. This research received partially support from the Global STEM Professorship Scheme from the Hong Kong Special Administrative Region.

## Impact Statement

This paper presents work whose goal is to advance the field of Machine Learning. There are many potential societal consequences of our work, none which we feel must be specifically highlighted here.

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

# A. Appendix

## A.1. Training and Testing Datasets

### A.1.1. TRAINING AND IN-DOMAIN TESTING DATASET

We utilize the Ego-IRGBench (Su et al., 2025b) dataset to train and evaluate the in-domain performance of our method. Ego-IRGBench comprises 20,504 egocentric images sourced from the HOI4D (Liu et al., 2022) dataset. Each egocentric RGB image is accompanied by a depth map and an interaction description detailing the specific interaction taking place. Additionally, multiple interaction-related queries are annotated for each image, with each query paired with an answer and a corresponding pixel-level grounding mask. In total, the dataset contains over 1.6 million queries, each with associated text and pixel-level response pairs. The dataset is divided into training, validation, and test sets in a 5:2:3 ratio, resulting in a training set of 10,249 images with 806,982 queries, a validation set of 4,094 images with 322,208 query-response-mask pairs, and a test set of 6,161 images with 485,174 query-response-mask pairs.

### A.1.2. OUT-OF-DISTRIBUTION TESTING DATASET

The scarcity of large-scale, unified benchmarks for egocentric interaction reasoning and grounding poses a significant challenge for evaluating model generalization. To rigorously assess the out-of-distribution (OOD) performance of our method, we employ the EgoHOS dataset (Zhang et al., 2022a) as an external validation set. EgoHOS provides 11,743 egocentric images with high-quality, per-pixel segmentation masks for hands and active objects. It is important to note that the annotations in EgoHOS are tailored for segmentation and thus only support the pixel grounding subtask. Consequently, our OOD evaluation specifically focuses on grounding performance, which constitutes the fundamental and most demanding objective of the Ego-IRG benchmark. To ensure the most comprehensive assessment of model robustness, we utilize the entire EgoHOS dataset, encompassing its official training, in-domain, and out-of-domain subsets as a consolidated validation set for this purpose.

## A.2. Training Strategy

Our framework is trained in two sequential stages. Stage 1 performs supervised fine-tuning to obtain reliable analysis features, while Stage 2 optimizes the response module with reinforcement learning. In Stage 2, we adopt a stabilized variant of Group Relative Policy Optimization (GRPO), which estimates advantages by comparing multiple responses sampled from the same prompt, avoiding the need for an additional value network.

Given a prompt $q$, the old policy samples a group of responses $\{o_i\}_{i=1}^{G} \sim \pi_{\theta_{\text{old}}}(\cdot \mid q)$. Each response receives a scalar reward $r_i$, and its group-relative advantage is computed as

$$\hat{A}_i = \frac{r_i - \bar{r}}{\sigma_r + \epsilon}, \quad \bar{r} = \frac{1}{G} \sum_{j=1}^{G} r_j. \tag{9}$$

To improve optimization stability, we replace the vanilla symmetric clipped surrogate with an asymmetric clipped ratio inspired by DAPO (Yu et al., 2025). The token-level ratio is bounded by two independent thresholds:

$$\tilde{\rho}_{i,t} = \text{clip}\left(\rho_{i,t}, 1 - \epsilon_{\text{low}}, 1 + \epsilon_{\text{high}}\right), \quad \epsilon_{\text{high}} \gg \epsilon_{\text{low}}. \tag{10}$$

For compactness, we define the token-level group average as

$$\mathbb{E}_{i,t}[\phi_{i,t}] = \mathbb{E}_{q,\{o_i\}} \left[ \frac{1}{G} \sum_{i=1}^{G} \frac{1}{|o_i|} \sum_{t=1}^{|o_i|} \phi_{i,t} \right]. \tag{11}$$

The Stage-2 objective is then written as

$$\mathcal{L}_{\text{SGRPO}} = \mathbb{E}_{i,t} \left[ \tilde{\rho}_{i,t} \hat{A}_i - \beta D_{\text{KL}}\left(\pi_\theta^{i,t} \| \pi_{\text{ref}}^{i,t}\right) \right], \tag{12}$$

where $\pi_\theta^{i,t} = \pi_\theta(\cdot \mid q, o_{i,<t})$ and $\pi_{\text{ref}}$ is a frozen reference policy.

During reinforcement learning, the vision encoder $\mathcal{E}_v$, text encoder $\mathcal{E}'_t$, and mask generation model $\mathcal{G}_{sam}$ are kept frozen. The final reward combines format correctness, answer relevance, and grounding accuracy:

$$R = \lambda_f R_f + \lambda_a R_a + \lambda_g R_g. \tag{13}$$

Here, $R_f$ encourages valid `<answer>` and `<bbox>` output structure, $R_a$ measures the textual consistency between the predicted and ground-truth answers, and $R_g$ evaluates spatial grounding accuracy by mask IoU.

### A.3. Implementation Details

For all experiments, we adopt a unified instruction template, $[\text{INST}]\ \langle \text{Img} \rangle \langle \text{ImageHere} \rangle \langle / \text{Img} \rangle\ [\text{task}]\ \text{Instruction}\ [/\text{INST}]$, to structure input queries and analysis instructions.

Our implementation consists of two sequentially trained stages. The coarse-grained analysis stage is built upon Qwen2.5-VL-3B, while the fine-grained response stage is built upon Qwen2.5-VL-7B. The vision encoders are frozen in both stages. Stage 1 is trained with supervised fine-tuning to generate holistic interaction descriptions and reliable global interaction descriptors. Stage 2 is trained with Stabilized GRPO to generate query-conditioned textual answers and bounding boxes, while SAM2-Large is kept frozen and used only to convert predicted boxes into pixel-level masks.

All experiments are conducted on four NVIDIA A800 80GB GPUs with BF16 mixed precision and Flash Attention. For SFT, we use a global batch size of 8, with 2 samples per GPU. For the RL stage, the total rollout/optimization batch size is 32 with group size 4. The RL stage takes approximately 10 hours for 1,000 steps, while the complete two-stage training pipeline takes approximately 13 hours in total. We use the AdamW optimizer with a learning rate of $1 \times 10^{-5}$ and a weight decay of 0.05. All images are resized to $448 \times 448$ pixels and normalized using a mean of $[0.55, 0.55, 0.53]$ and a variance of $[0.22, 0.23, 0.25]$.

### A.4. Additional Ablation Study

In this section, we provide additional ablation studies to further analyze the design choices and robustness of EARL. Specifically, we examine the effect of different answer relevance rewards, evaluate the training stability across random seeds, and compare EARL with a separately trained multi-model pipeline. These experiments provide additional evidence for the effectiveness and stability of the proposed unified analysis-guided RL framework.

**Effect of Answer Relevance Reward.** To examine whether the lexical answer reward is overly sensitive to paraphrases, we replace the original ExactMatch + Levenshtein reward with an embedding-based Sentence-BERT (SBERT) similarity reward. As shown in Table 5, the original reward achieves better answering and grounding performance. We hypothesize that the ExactMatch + Levenshtein reward provides a more structured and discriminative optimization signal for Ego-IRG, while the embedding-based reward may over-smooth semantically similar but spatially different responses, thereby weakening the joint optimization of answering and grounding.

*Table 5.* **Ablation of answer relevance reward on Ego-IRGBench test set.**

| Reward Type | Answering M↑ | Answering C↑ | Grounding cIoU↑ |
|---|---|---|---|
| ExactMatch + Levenshtein | **0.939** | **6.682** | **65.48** |
| SBERT-based Similarity | 0.920 | 5.579 | 62.58 |

**Robustness Across Random Seeds.** We further evaluate the stability of EARL by repeating training with three different random seeds. As shown in Table 6, the variance across seeds is small for analysis and answering metrics, and the grounding cIoU remains consistently strong. These results indicate that the proposed analysis-guided RL training is stable with respect to random initialization.

*Table 6.* **Robustness across random seeds on Ego-IRGBench test set.**

| Seed | Ana. M↑ | Ana. C↑ | Ans. M↑ | Ans. C↑ | Gro. cIoU↑ |
|------|---------|---------|---------|---------|------------|
| 42 | 0.542 | 1.522 | 0.939 | 6.682 | 65.48 |
| 123 | 0.531 | 1.509 | 0.931 | 6.604 | 63.17 |
| 3407 | 0.546 | 1.524 | 0.947 | 6.663 | 67.53 |
| Mean ± Std | 0.540±0.008 | 1.518±0.008 | 0.939±0.008 | 6.650±0.040 | 65.39±2.18 |

**Comparison with Separately Trained Subtask Models.** We also compare EARL with a separately trained multi-model pipeline, where independent modules are used for analysis, answering, and grounding. Although such a design is technically feasible, it introduces computational redundancy and weakens cross-task consistency. As shown in Table 7, EARL achieves better performance across all metrics with fewer parameters and lower inference latency, demonstrating the benefit of unified analysis-guided optimization.

*Table 7.* **Comparison between separately trained subtask models and EARL on Ego-IRGBench test set.**

| Method | Params | Time | Ana. M↑ | Ana. C↑ | Ans. M↑ | Ans. C↑ | cIoU↑ |
|--------|--------|------|---------|---------|---------|---------|-------|
| Separate Models | ∼17.3B | 17.2s | 0.466 | 1.423 | 0.891 | 4.607 | 43.86 |
| EARL | ∼10B | 5.3s | **0.542** | **1.522** | **0.939** | **6.682** | **65.48** |

## A.5. Qualitative Evaluation

This section provides a qualitative comparison of our proposed EARL against the baseline method (Bai et al., 2025) and the ground truth, as illustrated in Figure 5 and Figure 6. To ensure a comprehensive evaluation, we select representative samples from the three distinct categories present in the original Ego-IRGBench dataset: no-target, single-target, and multi-target scenarios.

Figure 5 presents the qualitative results for the no-target and single-target samples. A primary observation is that the textual analysis generated by the baseline model is notably more verbose compared to the concise and accurate descriptions in the ground truth and those produced by our EARL. More critically, in terms of visual grounding, our method demonstrates superior performance in generating masks with precise boundaries, closely aligning with the ground truth annotations. This precision in delineating object contours is a clear advantage over the comparatively coarser masks generated by the baseline. Furthermore, EARL exhibits a more robust understanding of user queries. The generated outputs are highly relevant and directly responsive to the query's intent, a capability that is crucial for the adaptability of such systems to downstream applications with diverse and dynamic user instructions.

The effectiveness of our approach is further validated on more complex, multi-target samples, as shown in Figure 6. Despite the increased complexity, the grounding masks produced by EARL remain highly accurate, effectively segmenting multiple interacting objects without significant performance degradation. This consistency across varying levels of scene complexity underscores the robustness and generalizability of our proposed method.

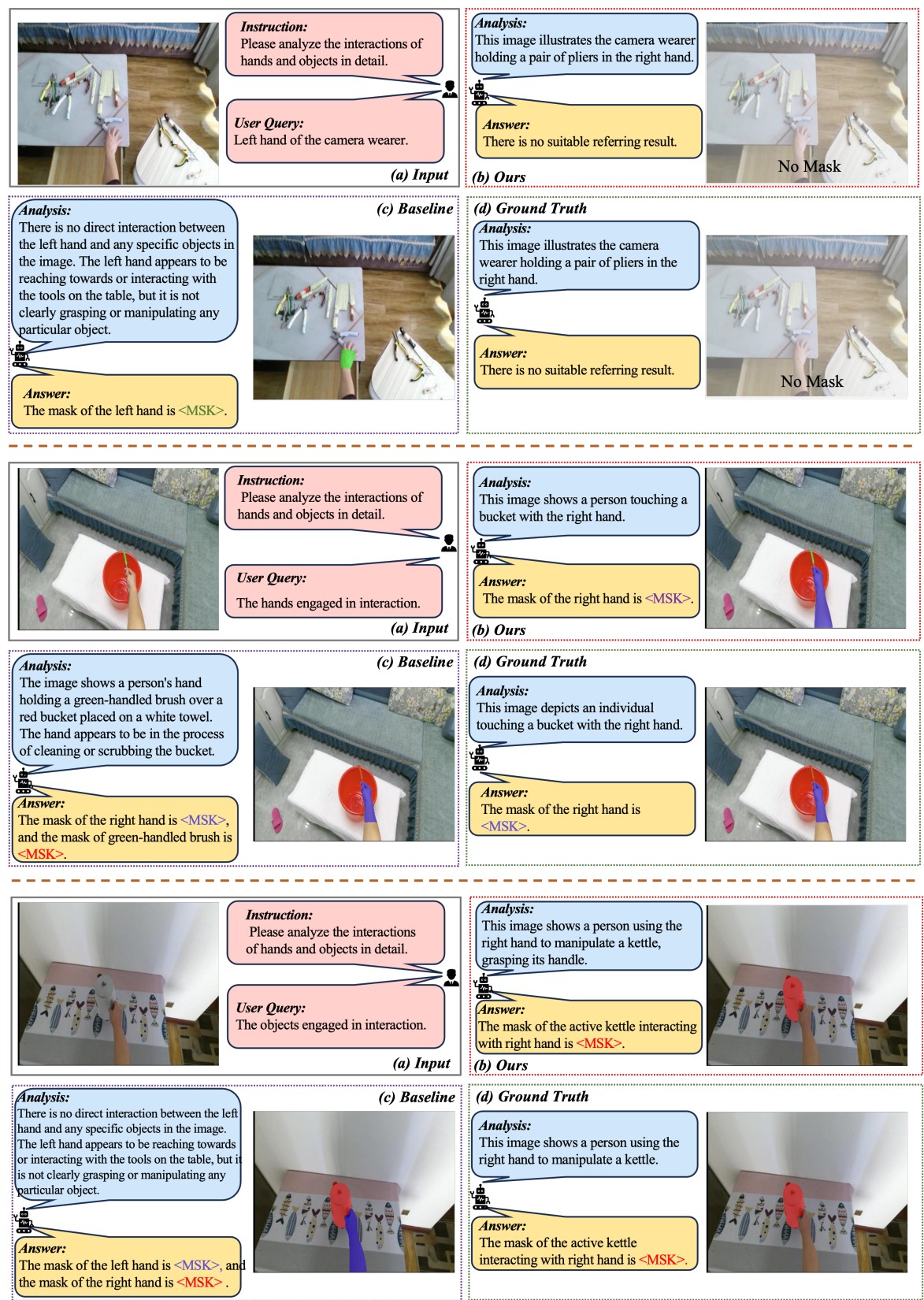

*Figure 5.* No-target and single-target visualization samples.

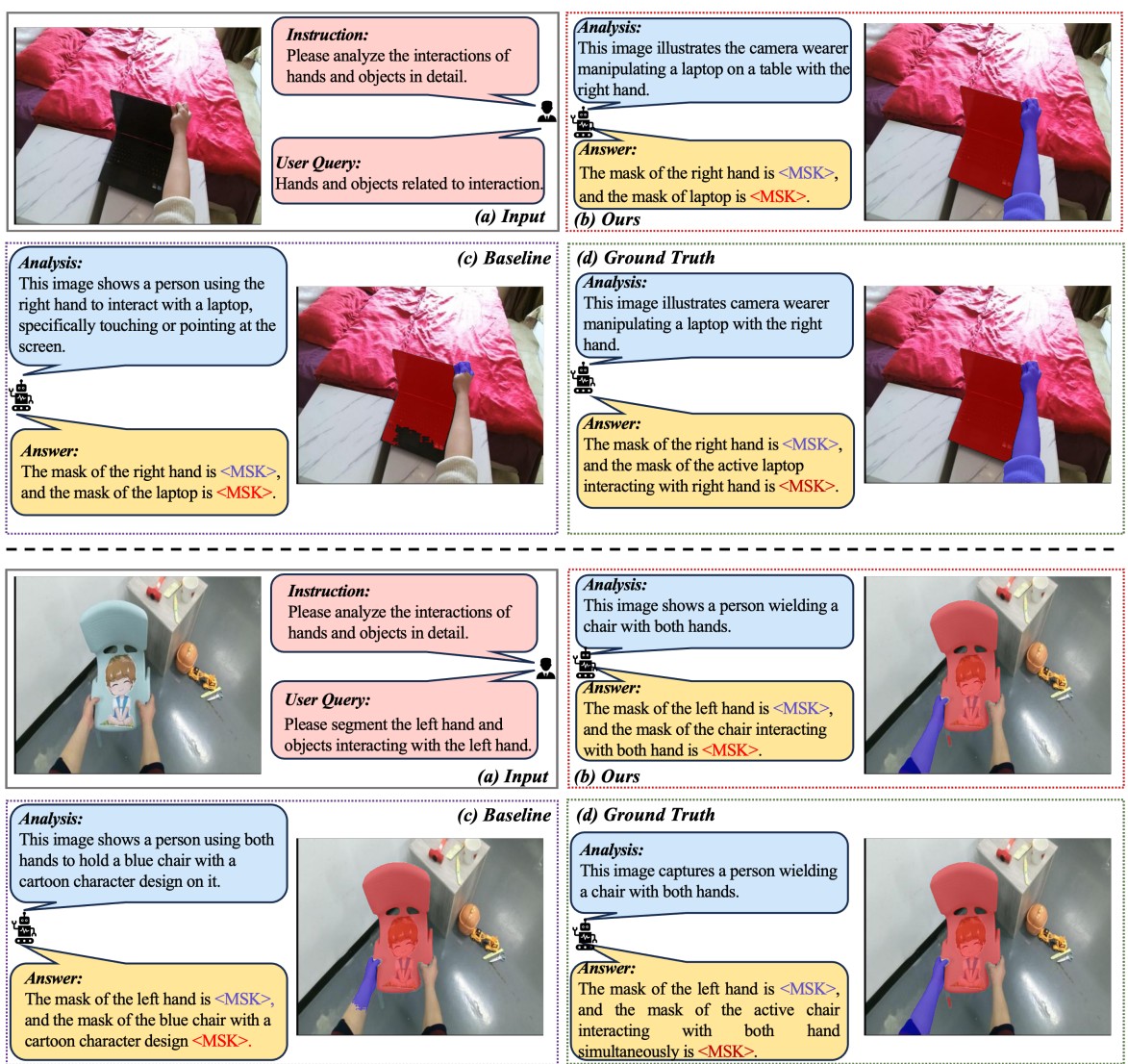

*Figure 6.* Multi-target visualization samples.

