# OpenReview forum: "EARL: Towards a Unified Analysis-Guided Reinforcement Learning Framework for Egocentric Interaction Reasoning and Pixel Grounding"
_ICML.cc/2026/Conference — ICML 2026 regular_

### Official Review · Reviewer_HvNk · 2026-03-02

**Soundness:** 3
**Presentation:** 3
**Significance:** 2
**Originality:** 2
**Overall Recommendation:** 3
**Confidence:** 3

**Summary:**

This paper proposes a two-stage framework for unified egocentric interaction reasoning and grounding. The authors propose coarse-grained interaction analysis, analysis-guided feature fusion, and task-specific reward to power GRPO. Empirical results show improvements in both in-domain grounding and OOD performance.

**Compliance With Llm Reviewing Policy:**

Affirmed.

**Final Justification:**

Thanks to the authors for their detailed response. Unfortunately, I am keeping my initial rating, as I still feel the contribution lacks sufficient novelty.

**Key Questions For Authors:**

1. The OOD dataset is only used for grounding evaluation. Can you provide the evidence that the unified reasoning pipeline generalizes to other task formulations (e.g., QA)?
2. Does the enhanced reasoning capability by EARL in the egocentric setting generalize to third-person benchmarks, or does it instead lead to performance degradation outside the egocentric domain?
3. It appears that the paper does not explicitly discuss its limitations. What do you consider to be the main limitations of this study?

**Limitations:**

The paper does not discuss its limitations. A discussion of the failure cases would add more depth to the paper.

**Strengths And Weaknesses:**

Strengths
1. The coarse-to-fine two-stage design is well structured, and the global interaction descriptor is clearly defined and consistently integrated into the second stage to improve performance.
2. The method demonstrates improvements in grounding accuracy compared to several RL-based and non-RL baselines.

Weaknesses
1. While the paper combines holistic analysis, query answering, and segmentation into a unified framework, it does not provide direct evidence that jointly optimizing is necessary to optimizing each task independently. A more thorough comparison, such as training separate RL models for each subtask and reporting their performance as well as training and inference costs, would strengthen the claim of the whole framework.
2. The novelty appears moderate: the RL component largely follows standard optimization, and the primary contribution lies in the multi-stage pipeline and feature fusion strategy tailored to this multi-task setting.

---

> ### Author Rebuttal · Authors · 2026-03-31
>
> **W1: Separate models.**
>
> **A1:**
>
> Thank you for your valuable feedback.
>
> Technically, training independent RL models for the three sub-tasks is indeed feasible, but this approach has several key drawbacks:
>
> 1) Computational and memory inefficiency. Three independent models require separate encoding of the same input, leading to unnecessary computational redundancy and increased memory consumption.
>
> 2) Limited performance and consistency. The absence of cross-task information sharing causes overfitting and ignores subtask correlations. Moreover, separate training breaks the alignment between textual answers and pixel masks, resulting in inconsistent outputs.
>
> We present additional results below:
>
> | Method | Params | Inference Time (s/sample) | Analysis M | Analysis C | Answering M | Answering C | Grounding cloU |
> |-|-|-|-|-|-|-|-|
> | Separate | ～17.3B  | 17.2 | 0.466 | 1.423 | 0.891 | 4.607 | 43.86 |
> | EARL (Ours) | ～10B | 5.3 | 0.542 | 1.522 | 0.939 | 6.682 | 65.48|
>
> The parameters of the individual modules is computed as: 7B for analysis, 3B for answering, and ~7.3B for grounding. Results show our unified framework achieves better performance across all metrics with fewer parameters and faster inference, validating the effectiveness of joint optimization.
>
> **W2: Novelty.**
>
> **A2:**
>
> Thank you for your feedback. We believe a paper's contribution is not only in introducing entirely new concepts, but also in how each component are innovatively integrated and deeply adapted to complex tasks. For the challenging task of Ego-IRG, our contribution is not a mere aggregation of existing methods. Rather, we make critical design choices at both the task and methodological levels, enabling seamless synergy among modules.
>
> Specifically, our methodological contributions are threefold:
>
> 1) The Ego-IRG task involves three multimodal subtasks, and unifying reinforcement learning for such a multi-output setting is highly challenging. Our two-stage EARL framework directly tackles this core issue.
>
> 2) We introduce the AFS, a novel component that implements a "filter-and-fusion" paradigm, using coarse-grained analysis features to guide fine-grained response generation (see A2 to reviewer #XBSG).
>
> 3) As reward design is critical in RL, we propose a tailored multi-faceted reward mechanism for the unified Ego-IRG task, representing a key methodological innovation.
>
> Experimental results demonstrate that our method consistently outperforms both existing RL-based approaches and egocentric-specific methods, firmly validating the novelty and effectiveness of our proposed methodology.
>
>
> **Q1: OOD inference.**
>
> **A3:** Please refer to A2 in the answer to the reviewer #XBSG.
>
> **Q2: On the third-person domain.**
>
> **A4:**
>
> Thank you for this important question.
>
> We clarify that our method is **specialized for the Ego IRG task**, not for cross-domain generalization. Egocentric has unique domain traits (large background variation, prominent hands), distinct from third-person scenarios. Thus, we do not aim for a general-purpose architecture, but focus on maximizing performance in the egocentric domain.
>
> We explain why direct evaluation on third-person benchmarks is unsuitable:
>
> 1. Task mismatch: Our task mainly focuses on hand-environment interaction, which is fundamentally different from the objectives of third-person benchmarks. These differences in task purpose and structure make direct transfer evaluation less comparable.
>
> 2. Domain specificity: Our method leverages first-person hand–object spatial cues, so performance may degrade in third-person scenarios due to domain shift. Owing to rebuttal time constraints, we cannot perform thorough comparative experiments.
>
> In summary, our method achieves significant performance gains in egocentric scenarios, with its design and optimization tailored to this domain.
>
> Furthermore, we will add "domain generalization" as an important direction for future work (see A5).
>
> **Q3: Limitations.**
>
> **A5:**
>
> Thank you for this suggestion. Our limitations are:
>
> 1. Limited small-object grounding: Our model struggles to parse small or occluded interactive objects. We will show representative samples in the "Failure Case" section.
>
> 2. Domain generalization: Our method is designed and optimized specifically for egocentric scenarios. As a result, the model may degrade in performance when applied to a third-person view.
>
> To address these limitations, Future work will include:
>
> 1) Designing fine-grained grounding with multi-scale fusion or iterative refinement to improve small-object grounding.
>
> 2) Exploring unified architectures that work well for both egocentric and exocentric.
>
> We will add these discussions to the revised manuscript.

---

> > ### Author Rebuttal · Reviewer_HvNk · 2026-04-03
> >
> > Thank you for the detailed response. However, I still feel that my concerns have not been fully resolved. In my view, the main contributions appear to lie more in engineering integration and system-building rather than methodological innovation, which may be less aligned with the expectations for ICML.

---

> > > ### Author Response · Authors · 2026-04-07
> > >
> > > We sincerely thank the reviewer for your continued engagement with our work and for the thoughtful follow-up comments.
> > >
> > > We would like to clarify that our contribution is not simply the combination of several known components, but **a new reasoning-and-learning formulation for comprehensive egocentric interaction understanding**.  Importantly, each of our contributions is designed to address a **specific, identified problem**, rather than being a simple stacking of existing techniques.
> > >
> > > First, existing MLLM methods that support multiple output modalities typically relate different subtasks **implicitly**, e.g., through shared representations, modality-specific heads, and joint end-to-end training. Under such a paradigm, analysis, answering, and grounding are still largely treated as parallel objectives. In contrast, we observed that **these subtasks are inherently correlated**: the coarse global semantics generated by analysis can serve as informative prior knowledge for subsequent fine-grained answering and grounding. Based on this observation, we proposed a two-stage coarse-to-fine framework, which **explicitly models cross-subtask dependency through stage-wise knowledge transfer**. In our view, the novelty lies not in the generic phrase “coarse-to-fine,” but in **introducing an explicit subtask-collaborative formulation** for egocentric multimodal reasoning.
> > >
> > >
> > > Second, our subtask-collaborative framework introduces a potential problem: errors from the analysis stage may accumulate and propagate to subsequent subtasks. To address this, we propose **AFS**, which is not a generic fusion block but designed for the case where the first-stage analysis provides a useful yet noisy semantic prior. Its role is to **adaptively filter task-irrelevant information before fusion**, implementing a filter-then-fuse mechanism tailored to two-stage reasoning. This design is methodologically motivated by the noisy-prior problem induced by our framework, rather than being an off-the-shelf engineering component.
> > >
> > >
> > > Third, our RL design is likewise not a standard application. Jointly optimizing heterogeneous outputs raises non-trivial challenges in reward shaping and credit assignment. To address this, we design a **task-specific multi-faceted reward mechanism** for the Ego-IRG setting, enabling unified optimization across interdependent subtasks.
> > >
> > >
> > > Therefore, we respectfully believe the paper goes beyond system-building: it proposes:
> > >
> > > (i) a new **explicit subtask-collaborative coarse-to-fine formulation**,
> > >
> > > (ii) a new mechanism for **exploiting noisy coarse-grained semantic priors**, and
> > >
> > > (iii) a new **RL-based unified optimization strategy** for heterogeneous outputs.
> > >
> > >
> > > We thank the reviewer for pointing out that the current presentation may make the work appear more engineering-oriented than intended. To address this, in the final version we will substantially revise the writing to foreground the methodological novelty. Specifically, we will (i) more explicitly position our work as a **new subtask-collaborative coarse-to-fine formulation** for egocentric multi-output reasoning, (ii) clarify that AFS is a **task-specific noisy-prior filtering mechanism** rather than a generic fusion module, and (iii) provide a more detailed explanation of the **RL reward design** and credit assignment challenges for heterogeneous outputs. We will also strengthen the empirical section with clearer ablations and discussion to better isolate the contribution of each methodological component.
> > >
> > > We respectfully hope that these clarifications and planned revisions further address the reviewer's concerns.

---

### Official Review · Reviewer_PuYT · 2026-03-08

**Soundness:** 3
**Presentation:** 3
**Significance:** 3
**Originality:** 3
**Overall Recommendation:** 4
**Confidence:** 3

**Summary:**

1. The paper formalizes Ego-IRG: given an egocentric image and a user query, it aims to generate a holistic interaction description, a query-specific textual answer, and pixel-level grounding masks.

2. It proposes EARL, a two-stage coarse-to-fine framework that (i) extracts a global interaction descriptor from coarse analysis and fuses it via an Analysis-guided Feature Synthesizer (AFS), and (ii) trains the fine stage with GRPO using rewards for format, answer relevance, and grounding IoU, reporting improved grounding on Ego-IRGBench and OOD EgoHOS.

**Compliance With Llm Reviewing Policy:**

Affirmed.

**Key Questions For Authors:**

No questions from my side.

**Limitations:**

yes

**Strengths And Weaknesses:**

# Strengths

1. The method is specified cleanly end-to-end (two-stage pipeline + AFS fusion + GRPO rewards), with concrete equations and diagrams that make the workflow easy to follow.

2. The paper targets a practically relevant unified Ego-IRG setting and shows strong grounding gains in-domain and OOD, with ablations supporting the role of AFS.

# Weaknesses

1. The answer reward uses ExactMatch + Levenshtein ratio, which may be brittle to paraphrases; a small ablation with an embedding-based similarity (or a short robustness check on paraphrased answers) would strengthen this component.

2. Table 1 lists strong open-source baselines (e.g., Qwen3-VL, Seg-R1), but it is unclear whether these models were fine-tuned on Ego-IRGBench; please state this explicitly and, ideally, align the comparison setting (zero-shot vs. fine-tuned) across all methods.

3. The comparison methods should include recent SOTA pixel-level MLLMs (e.g., SA2VA, UniPixel).

---

> ### Author Rebuttal · Authors · 2026-03-31
>
> **W1:** The answer reward uses ExactMatch + Levenshtein ratio, which may be brittle to paraphrases; a small ablation with an embedding-based similarity (or a short robustness check on paraphrased answers) would strengthen this component.
>
> **A1:**
>
> Thank you for your valuable suggestion. We conducted an additional experiment by replacing the original reward (ExactMatch + Levenshtein ratio) with an **embedding-based similarity metric** using **Sentence-BERT (SBERT)** to evaluate answer quality.
>
> The results on the Ego-IRGBench test set are summarized below:
>
> | Reward | Answering (M) | Answering (C) | Grounding (cloU) |
> |--|--|--|--|
> | Original (ExactMatch + Levenshtein) | 0.939 | 6.682 | 65.48 |
> | SBERT-based | 0.920 | 5.579 | 62.58 |
>
> As shown, the original reward consistently outperforms the SBERT based counterpart across all metrics. We hypothesize that the ExactMatch + Levenshtein combination provides a more structured and stable training signal, whereas the embedding based reward may introduce undesired smoothing that hinders precise optimization for both answer generation and grounding. Notably, grounding performance also drops by nearly 3% cloU, indicating that the choice of answer reward indirectly affects the joint learning of the policy across tasks.
>
> We will add this ablation study and a brief discussion to the revised manuscript to clarify the rationale behind our reward design.
>
> **W2:** Table 1 lists strong open-source baselines (e.g., Qwen3-VL, Seg-R1), but it is unclear whether these models were fine-tuned on Ego-IRGBench; please state this explicitly and, ideally, align the comparison setting (zero-shot vs. fine-tuned) across all methods.
>
> **A2:**
>
> We thank the reviewer for this important clarification, and we apologize for not making the evaluation settings clear in the original manuscript.
>
> In Table 1, different comparison methods adopt different training strategies based on their characteristics:
>
> Egocentric-specific methods (e.g., ANNEXE) are fine-tuned on Ego-IRGBench using supervised fine-tuning (SFT). This setting best reflects the performance of task-specific models designed for egocentric interaction understanding.
>
> For general-domain methods (including non-RL models such as QwenVL and InternVL, as well as RL-based models such as Seg-R1 and Seg-Zero), we choose to evaluate them in a zero-shot manner. This choice follows the common practice in existing benchmark evaluations, where general-domain models are typically reported in a zero-shot setting to demonstrate their out-of-the-box generalization capability, while task-specific models are fine-tuned to show their upper-bound performance.
>
> We will explicitly indicate these evaluation settings in the revised manuscript in the Appendix. We thank the reviewer for helping us improve the clarity of our comparisons.
>
> **W3:** The comparison methods should include recent SOTA pixel-level MLLMs (e.g., SA2VA, UniPixel).
>
> **A3:**
>
> We thank the reviewer for this valuable suggestion. Following the reviewer's recommendation, we conducted additional experiments comparing our method with SA2VA (InternVL3-8B and 8B variants) and UniPixel-7B on both the Ego-IRG test/val sets and the out-of-distribution EgoHOS dataset.
>
> The results are summarized below:
>
> 1) Ego-IRG Test Set:
>
> | Method | Analysis (M) | Analysis (C) | Answering (M) | Answering (C) | Grounding (cloU) |
> |--|--|--|--|--|--|
> | Sa2VA-InternVL3-8B | 0.207 | 0.192 | 0.486 | 2.309 | 31.17 |
> | Sa2VA-8B | 0.188 | 0.115 | 0.754 | 2.656 | 32.69 |
> | UniPixel-7B | 0.165 | 0.075 | 0.158 | 1.136 | 14.01 |
> | EARL (Ours) | **0.542** | **1.522** | **0.939** | **6.682** | **65.48** |
>
> 2) Ego-IRG Val Set:
>
> | Method | Analysis (M) | Analysis (C) | Answering (M) | Answering (C) | Grounding (cloU) |
> |--|--|--|--|--|--|
> | Sa2VA-InternVL3-8B | 0.280 | 0.198 | 0.489 | 2.320 | 37.97 |
> | Sa2VA-8B | 0.189 | 0.120 | 0.448 | 2.644 | 34.89 |
> | UniPixel-7B | 0.168 | 0.065 | 0.155 | 1.128 | 13.63 |
> | EARL (Ours) | **0.542** | **1.522** | **0.939** | **6.682** | **65.48** |
>
> 3) OOD EgoHOS dataset:
>
> | Method | Left Hand | Right Hand | Left-hand Object | Right-hand Object | Two-hand Object | Overall |
> |--|--|--|--|--|--|--|
> | Sa2VA-8B | 48.56 | 45.82 | 26.91 | 26.77 | 37.04 | 37.63 |
> | Sa2VA-InternVL3-8B | 41.24 | 41.47 | 25.95 | 26.54 | 37.45 | 34.08 |
> | UniPixel-7B | 28.54 | 24.82 | 11.12 | 12.45 | 19.06 | 21.14 |
> | EARL (Ours) | **52.30** | **62.44** | 19.10 | 20.57 | 22.85 | **38.21** |
>
> As shown, our method substantially outperforms SA2VA and UniPixel across nearly all metrics on the in-distribution Ego-IRG test and val sets. On the OOD EgoHOS dataset, our method achieves the best overall performance (38.21) and particularly strong results on hand detection, demonstrating superior generalization to unseen scenarios.
>
> We will include these comparisons in the revised manuscript. We thank the reviewer for helping us strengthen the empirical evaluation of our work.

---

### Official Review · Reviewer_UMca · 2026-03-09

**Soundness:** 2
**Presentation:** 3
**Significance:** 3
**Originality:** 2
**Overall Recommendation:** 4
**Confidence:** 3

**Summary:**

The paper proposes EARL, a two-stage, RL-enhanced framework for egocentric interaction reasoning and pixel-level grounding. Stage 1 produces a holistic textual scene analysis and a global interaction descriptor; Stage 2 fuses this descriptor with image and query features via an Analysis-guided Feature Synthesizer (AFS), predicts query-aligned answers and bounding boxes, and uses a frozen SAM2 to generate masks. The second stage is optimized with GRPO using a composite reward over format correctness, answer relevance, and grounding accuracy, yielding strong gains on Ego-IRGBench (notably +8.37% cIoU over prior RL-based SOTA) and improved OOD grounding on EgoHOS.

**Compliance With Llm Reviewing Policy:**

Affirmed.

**Key Questions For Authors:**

1.AFS clarification: How exactly are Q/K/V computed and which tensors feed them? Do any of Q/K/V derive from F_emb (image+query) or only from F_ana? Please provide corrected equations and a brief schematic, and explain how analysis truly “guides” fusion beyond a residual addition.

2.GRPO details: What is the group size, sampling temperature/top-p, KL penalty to the SFT reference (if any), baseline computation, and whether you normalize or clip rewards/advantages? Any measures used to stabilize training (entropy bonuses, reward scaling)?

3.Efficiency: Please report total RL training time, GPU hours, and inference latency, and compare with top baselines to assess practicality.

**Limitations:**

Yes

**Strengths And Weaknesses:**

Strengths：

1.The paper presents a coherent coarse-to-fine design that explicitly transfers analysis-derived global semantics into query-driven response and grounding. The AFS module formalizes an analysis-guided fusion step and provides a mechanism to inject scene-level priors into instance-level grounding. Applying GRPO to unify textual reasoning and spatial grounding with a multi-part reward is well-motivated and practical for non-differentiable components (SAM2).

2.Comprehensive in-domain evaluation on Ego-IRGBench with both non-RL and RL MLLM baselines; consistent, sizable gains on the principal grounding metric (cIoU).

3.The overall system pipeline is easy to follow; task definition, training setup, and evaluation metrics are clearly stated at a high level. The reward components are conceptually simple and help structure the RL objective around end-task desiderata.

Weaknesses：

1.Details of GRPO usage are thin (group size, sampling temperature, KL/entropy regularization, baseline computation, reward normalization), making it difficult to assess stability and reproducibility.

2.The AFS description is ambiguous: equations indicate Q/K/V are derived only from the analysis descriptor, and F_emb is added post-attention, suggesting limited or no cross-attention between analysis and (image, text) features; this raises concerns about whether AFS truly fuses modalities or just adds a residual.

3.Fairness of comparisons: EARL relies on a strong frozen SAM2 for mask generation; it is unclear whether baseline RL methods also have access to the same mask generator or comparable post-processing. A control without SAM2 or with a uniform generator across methods would better isolate the policy quality.

4.Ego-IRGBench includes depth, but the method uses only RGB; an ablation with depth or justification would be informative.

---

> ### Author Rebuttal · Authors · 2026-03-31
>
> **W3: Fairness of SAM2.**
>
> **A1:**
>
> We thank the reviewer for this concern regarding fairness.
>
> First, in our framework, the MLLM predicts bounding boxes, while SAM2 is frozen and only converts boxes to masks. Since SAM2 is fixed, strong grounding performance comes from accurate bounding box predictions, not from SAM2 itself.
>
> Second, using SAM2 as a mask generator is a common practice in recent MLLM-based grounding literature. For fairness, we apply the same frozen SAM2 to all methods in Table 1, including RL-based baselines and bounding-box-only non-RL approaches. Therefore, under this unified and consistent setting, our comparison is fair.
>
> We acknowledge the insufficient description and will clarify this in the revised manuscript.
>
> **W4: Depth usage.**
>
> **A2:**
>
> Thank you for this suggestion.
>
> In Table 1, some comparison methods, such as ANNEXE, explicitly leverage depth as additional information to assist in recognizing interacting hands and objects. However, our RGB-only framework already substantially outperforms existing RGB-D methods by a large margin. This demonstrates that our RL-based framework provides sufficiently strong feature extraction and reasoning capabilities, making depth auxiliary unnecessary.
>
> Additionally, incorporating depth would require extra encoders and fusion modules, increasing model complexity and training difficulty. By using only RGB, our method maintains a more efficient architecture while achieving SOTA performance.
>
> **Q1 & W2: AFS clarification.**
>
> **A3:**
>
> Thank you to the reviewers for raising this important question. We understand the concerns about AFS and would like to clarify its design rationale and guiding mechanism in detail.
>
> The design rationale for AFS can be found in A4 of the response to reviewer #XBSG. In conclusion, AFS follows a "selection-and-fusion" paradigm:
>
> 1) Selection: Instead of using cross-attention, we first apply self-attention to the analytical features to adaptively suppress noise and to enhance the information that is reliable and beneficial for subsequent tasks.
>
> 2) Fusion: The refined analytical feature is then injected as a semantic prior into the answering and grounding subtasks via direct addition with the multimodal features.
>
> Thus, the Q, K, and V are all derived from the analytical feature $F_{ana}$ to perform self-attention. The refined analytical feature is then directly added to the multimodal feature, as shown in Equation 4-5 on page 5 of the manuscript.
>
> We will clarify AFS’s role and guiding mechanism more explicitly in the revised manuscript.
>
> **Q2 & W1: GRPO details.**
>
> **A4:**
>
> Thank you for the detailed questions regarding our GRPO implementation. We provide the following specifics:
>
> - Group size: 4
>
> - Sampling: We use a sampling temperature of 1.0 and top‑p = 1.0 for exploration during rollout.
>
> - KL penalty: A KL penalty coefficient (`beta`) of 0.01 is applied to constrain the policy from deviating too far from the SFT reference model.
>
> - Baseline computation: We follow the standard GRPO formulation, where the baseline is computed as the mean reward within each group.
>
> - Reward/advantage normalization and clipping: We adopt the Stabilized GRPO variant from the Hugging Face TRL library. Instead of a single clipping ratio, we use two independent parameters: `ε_low` and `ε_high`. Specifically, we set a smaller `ε_low` and a significantly larger `ε_high`, which allows low‑probability (exploratory) tokens to have more room to increase, effectively preventing entropy collapse. No additional normalization or clipping is applied to rewards or advantages.
>
> - Training stabilization: Beyond the asymmetric clipping described above, we do not employ additional measures such as entropy bonuses or reward scaling, as the Stabilized GRPO formulation proved sufficient for stable training in our setting.
>
> We will add a brief discussion of these implementation details in the revised manuscript to improve reproducibility.
>
>
> **Q3: Efficiency:**
>
> **A5:**
>
> Thank you for the efficiency question. We emphasize that our primary focus is not on optimizing efficiency but on leveraging reinforcement learning to address the complex Ego-IRG task, thereby enhancing model performance and generalization.
>
> Nevertheless, we provide the following efficiency metrics for transparency.
>
> Training Details:
>
> - RL Training time: Approximately 10 hours for 1,000 steps
>
> - GPU hours: 40 GPU hours (4 A800 GPUs × 10 hours)
>
> - Total batch size: 32
>
> Inference Latency:
>
> - Our method: Approximately 5.2 seconds per sample on the test set
>
> - Seg-R1-7B: Approximately 4.0 seconds per sample on the test set
>
> As shown, our method incurs a moderate increase in inference time compared to Seg-R1-7B. This trade‑off is acceptable given the substantial performance gains we achieve across analysis, answering, and grounding tasks, as reported in our main results.

---

> > ### Author Rebuttal · Reviewer_UMca · 2026-04-04
> >
> > Thank you for the response. The author solved most of my questions.

---

### Official Review · Reviewer_XBSG · 2026-03-13

**Soundness:** 3
**Presentation:** 3
**Significance:** 2
**Originality:** 3
**Overall Recommendation:** 4
**Confidence:** 3

**Summary:**

This paper presents EARL, a two-stage framework for egocentric interaction reasoning and pixel grounding. The core idea is to first generate a coarse-grained textual analysis of hand-object-environment interactions, then use the resulting global interaction descriptor as a semantic prior to guide query-conditioned answer generation and pixel-level grounding. To improve the second stage, the authors introduce an Analysis-guided Feature Synthesizer (AFS) and optimize the model with GRPO using rewards for output format, answer relevance, and grounding accuracy. Experiments on Ego-IRGBench show strong gains, especially in grounding, and OOD results on EgoHOS suggest some degree of generalization. Overall, the paper addresses a meaningful application problem and the system appears practically effective, but the methodological novelty is moderate: much of the design builds on existing coarse-to-fine reasoning, intermediate-analysis conditioning, and RL-based reward shaping for segmentation and multimodal reasoning.

**Compliance With Llm Reviewing Policy:**

Affirmed.

**Final Justification:**

My previous concerns have been adequately addressed in multi-round discussion. I would like to raise my rating to weak accept (4).

**Key Questions For Authors:**

1. How much of the improvement comes from RL, as opposed to the two-stage analysis-guided design?
2. Why is AFS necessary and meaningfully different from a standard fusion module? Although AFS is presented as a key contribution, its current formulation appears close to a feature fusion block built from projection, reshaping, attention-style interaction, and residual addition. The ablation in Table 3 shows that it outperforms simpler fusion variants, but this alone does not fully explain why this specific design is needed.
3. Is the first-stage analysis reliable enough to serve as a semantic prior? The method relies on the assumption that the coarse-grained analysis produces trustworthy global interaction descriptors, but the analysis results themselves are not consistently dominant in the main table.
4. The main comparison mixes egocentric-specific models, general-domain models, non-RL methods, and RL-based methods, but the paper does not provide a sufficiently transparent summary of backbone size, trainable parameters, RL usage, external priors, or training budget across baselines.
5. How robust are the reported results across random seeds and reward settings, please provide more evidence.  Since the method relies heavily on RL, reporting only single-run numbers is not fully convincing. In particular, Table 4 suggests that changing the reward weights can noticeably alter performance, and one non-default configuration even gives higher grounding cIoU 69.87 than the default setting. The paper would be much stronger if it reported multi-seed results, variance, and a clearer explanation of the final hyper-parameter choice.

**Limitations:**

While the paper shows promising empirical gains, its main limitation is that the contribution appears largely task-specific and integrative rather than methodologically fundamental. The core design combines several existing ideas—coarse-to-fine reasoning, intermediate analysis as semantic prior, and RL-based reward shaping—in the egocentric interaction setting, but does not yet establish a broadly new learning principle or transferable modeling framework beyond this particular problem formulation. In addition, the framework depends on a relatively heavy multi-component pipeline, including separate analysis and answering decoders as well as an external mask generator, which may limit its conceptual elegance and practical scalability. More importantly, the current evidence for generalization remains narrow, as it is mainly demonstrated through OOD grounding rather than full-task transfer. As a result, even if the reported results are strong, the paper’s broader impact on the core machine learning community may remain somewhat limited.

**Strengths And Weaknesses:**

### Pros
+ The paper addresses a meaningful egocentric vision problem with clear practical relevance. Unifying scene-level interaction analysis, query-conditioned answering, and pixel grounding is useful for assistive agents and embodied systems operating in first-person environments.
+ The proposed pipeline is technically coherent and easy to follow. The coarse-to-fine design is intuitive: global interaction analysis is first extracted, then reused as semantic prior for downstream answering and grounding.
+ The empirical gains on grounding are substantial. The reported improvement on Ego-IRGBench, especially the large cIoU margin over prior RL-based methods, is the strongest part of the paper and suggests the method is practically effective.
+ The paper includes multiple forms of evaluation. In-domain comparison, OOD grounding evaluation, ablation studies, and qualitative results together provide a reasonably complete empirical picture.

### Cons
- The methodological novelty is moderate. The main ingredients—coarse-to-fine reasoning, intermediate analysis as prior, and RL-based reward shaping—are all closely related to existing ideas. The paper’s contribution appears more integrative than fundamentally novel.
- The AFS module does not appear sufficiently distinctive. Although presented as a key component, it functions largely as a feature fusion block, and the current evidence does not fully establish why this specific design is uniquely necessary.
- The generalization claim is somewhat overstated. The OOD evidence is limited to grounding on EgoHOS, without comparable OOD evaluation for analysis or answer generation. This supports some transferability, but not yet broad generalization of the full framework.
- The empirical robustness could be stronger. The paper does not provide enough evidence on variance across runs, sensitivity to training settings, or fairness of comparison under matched model scale and compute, which weakens confidence in the reported gains.

---

> ### Author Rebuttal · Authors · 2026-03-31
>
> **W1: Novelty.**
>
> **A1:** Please see A2 in response to reviewer #HvNK.
>
>
> **W3: OOD inference.**
>
> **A2:**
>
> Thank you for your valuable feedback. We explain why our OOD evaluation focused only on grounding:
>
> 1. Dataset limitations. Ego-IRG specifically targets structured human–object interaction understanding. Existing egocentric Q&A/captioning datasets often include interaction-irrelevant content, which mismatches our task definition.
>
> 2. Importance of grounding. Pixel-level grounding is the most challenging subtask and strongly reflects the model’s spatial understanding and generalization. Improvements in OOD grounding thus demonstrate the model’s core generalization ability.
>
> We will clarify the scope and limitations of our OOD evaluation in the revised manuscript.
>
> **Q1: RL improvement.**
>
> **A3:**
>
> Thank you for the question. We clarify the contribution of RL versus the two-stage design as follows:
>
> Single-stage SFT vs. RL: Single-stage RL (row 2) achieves large gains over single-stage SFT (row 1) across all metrics in Table 3.
>
> Two-stage SFT vs. RL: We add a two-stage SFT baseline with AFS fusion (SFT+AFS) below. Comparing it with RL+AFS, RL still yields consistent and significant gains under the same two-stage architecture.
>
> | Train     | Ana.M | Ana.C | Ans.M | Ans.C | Gro.cIoU |
> |-|-|-|-|-|-|
> | SFT + AFS | 0.540 | 1.457 | 0.498 | 4.313 | 43.85    |
> | RL + AFS  | 0.542 | 1.522 | 0.939 | 6.682 | 65.48    |
>
> These results confirm that RL provides substantial gains beyond the two-stage design. We will add the two-stage SFT results to the table and clarify RL's role in the revision.
>
> **Q2&W2: AFS.**
>
> **A4:**
>
> Thank you for this comment. We clarify the necessity and advantages of AFS:
>
> Necessity of AFS. In our framework, coarse-grained analysis provides structured, interpretable semantics of active entities and their spatial relations. Rather than a generic feature, it acts as a task-aligned semantic prior for answering and grounding. AFS is designed to exploit this structured information explicitly.
>
> AFS vs. Standard Fusion. Standard fusion always mixes features indiscriminately without filtering noise. However, AFS follows a selection-and-fusion paradigm:
>
> 1) Selection: AFS uses self-attention to highlight analysis features relevant to answering and grounding, while suppressing noise from incorrect predictions.
>
> 2) Fusion: The refined analysis is added to multimodal features, preserving fine-grained details while injecting task-related context. Table 3 shows that AFS outperforms simple fusion due to this selective mechanism.
>
> We further compare AFS with cross-attention fusion (CAF):
>
> | Method | Ana.M | Ana.C | Ans.M | Ans.C | Gro.cIoU |
> |-|-|-|-|-|-|
> | CAF| 0.502 | 1.487 | 0.790 | 5.241            | 52.36          |
> | AFS | 0.542      | 1.522      | 0.939       | 6.682       | 65.48          |
>
> Results show that AFS still achieves clear gains, demonstrating the effectiveness of the selection-and-fusion paradigm.
>
> **Q3: Analysis stage.**
>
> **A5:**
>
> Thank you for the question. We agree that first-stage analysis reliability is critical to our framework.
>
> Although analysis scores in Table 1 are not SOTA, this does not mean the semantic prior is unreliable. METEOR focuses on lexical matching, while our analysis prior focuses on capturing key interactive elements (active hand, object) rather than exact wording.
>
> We also recognize that the first-stage analysis is imperfect. So we propose the AFS to mitigate this via selective refinement: it enhances useful features and suppresses noise to avoid error propagation.
>
> Table 3 shows a large improvement from no fusion (39.81% cIoU) to AFS (65.48%), confirming that analysis provides effective guidance even with minor imperfections.
>
> We will incorporate the above analysis into the revision.
>
> **Q4: Comparisons.**
>
> **A6:**
>
> Thank you for this feedback. We have added a comprehensive table (see https://anonymous.4open.science/r/Rebuttal-8145/table.md) summarizing key aspects of all comparison methods. It will be included in the Appendix.
>
> **Q5&W4: Robustness.**
>
> **A7:**
>
> Thank you for the suggestion. We evaluate two different seeds:
>
> | Seed | Ana.M | Ana.C | Ans.M | Ans.C | Gro.cIoU |
> |-|-|-|-|-|-|
> | 42 (Ori)   | 0.542      | 1.522      | 0.939       | 6.682       | 65.48          |
> | 123  | 0.531      | 1.509      | 0.931       | 6.604       | 63.17          |
> | 3407  | 0.546      | 1.524      | 0.947       | 6.663       | 67.53          |
> | Mean & Std | 0.540±0.008 | 1.518±0.008  | 0.939±0.008  | 6.650±0.040  | 65.39±2.18  |
>
> Results show small variance across seeds, confirming that our framework is stable with respect to seed initialization.
>
> For hyperparameters, we have conducted ablation in Table 4 and discussed them in Section 4.2.4. Based on the results, we set \(\lambda_f\), \(\lambda_a\), and \(\lambda_g\) all to 1 to balance the subtasks equally and avoid bias toward any single task.
>
> We will add these results to the Appendix to show robustness.

---

> > ### Author Rebuttal · Reviewer_XBSG · 2026-04-04
> >
> > Thank you for the detailed rebuttal. The additional results help address some of my concerns, for example Q1, and Q5. However, several of my more important concerns are still only partially resolved:
> >
> > 1. Q2 / W2: the new comparisons show that AFS works better than several alternatives, but they still do not fully establish why this specific design is fundamentally necessary, rather than simply being a stronger fusion block in practice.
> >
> > 2. Q3: the rebuttal gives a reasonable explanation, but I still do not see sufficiently direct evidence that the analysis stage is consistently reliable enough to serve as a robust semantic prior.
> >
> > 3. W3: the clarification helps narrow the scope, but the OOD evidence is still limited to grounding, so the broader generalization claim for the full framework remains not fully supported.
> >
> > 4. Q4 / W4: the comparison setting is clearer now, but I would still prefer a more apples-to-apples discussion under more closely matched training regime, model scale, and compute conditions.
> >
> > Overall, the rebuttal strengthens the empirical side of the paper, but the key concerns above are not fully resolved.

---

> > > ### Author Response · Authors · 2026-04-08
> > >
> > > Q2 / W2: AFS.
> > >
> > > A1:
> > >
> > > We thank the reviewer for the continued engagement.
> > >
> > > We acknowledge that fusion is part of AFS's functionality, but not the primary function. The most important role of AFS is **selection**.
> > >
> > > In our two-stage framework, a straightforward way to use first-stage global information is to fuse it directly with multimodal features, but experiments with MLP, summation, and cross-attention show poor results. This is because direct fusion can **propagate first-stage errors or noise into fine-grained reasoning**.
> > >
> > > AFS is specifically designed to mitigate this problem through its select-and-fuse paradigm. Specifically, it adaptively filters the coarse global features using a self-attention mechanism before fusion, suppressing unreliable or noisy information while preserving useful global cues. As a result, AFS achieves the best performance.
> > >
> > > Thus, AFS is effective not because of strong fusion capacity, but because **its noise-filtering design is well-matched to our two-stage framework, where first-stage analysis may be imperfect.** This allows the model to use coarse global information without letting early errors dominate downstream reasoning.
> > >
> > > We respectfully hope that these clarifications further address the reviewer's concerns.
> > >
> > > Q3: analysis stage.
> > >
> > > A2:
> > >
> > > We thank the reviewer for the thoughtful follow-up.
> > >
> > > We would like to clarify that the analysis stage itself is not assumed to be perfectly reliable. Instead, the reliability comes from the **combination of coarse semantic priors and AFS filtering**, rather than from the analysis stage alone.
> > >
> > > Table 3 provides direct evidence:
> > >
> > > 1) Without using analysis features to guide downstream subtasks, performance is low (39.81% cIoU).
> > >
> > > 2) MLP and summation degrade performance, indicating that noise in the analysis features can harm downstream reasoning.
> > >
> > > 3) Simple fusion methods such as concatenation and cross-attention bring only marginal gains, suggesting that the features are useful but not effectively exploited.
> > >
> > > 4) AFS achieves the best result (65.48% cIoU), showing that its key advantage lies in filtering noisy information while preserving useful semantic cues.
> > >
> > > These results support our main point: the analysis stage alone is not robust enough, but it does provide valuable coarse semantic information. AFS makes this information reliable for downstream use by suppressing noise.
> > >
> > > We will make this distinction clearer in the revised manuscript.
> > >
> > > W3: OOD evaluation.
> > >
> > > A3:
> > >
> > > Thank you for the insightful comment.
> > >
> > > We acknowledge that our OOD evaluation is limited to the grounding. The main reason is that, to the best of our knowledge, there is **currently no suitable dataset** or established benchmark that supports OOD evaluation for answering and grounding in the egocentric interaction domain.
> > >
> > > To provide additional evidence, we will **add qualitative examples in the appendix showing model outputs (analysis, answering, grounding) on the EgoHOS dataset**. Although these results are not quantitative, they show that the model can still generate semantically reasonable textual outputs that are consistent with the predicted grounding results under OOD scenarios.
> > >
> > > Furthermore, in response to this comment, we will also revise the manuscript in two aspects: 1) **Acknowledge this limitation explicitly** in the Limitation section, clarifying that OOD evaluation is currently constrained to the grounding subtask due to benchmark availability. 2) **Add a Future Work direction** to construct a unified OOD benchmark that supports full task evaluation for egocentric interaction understanding.
> > >
> > > We hope these additions appropriately address the reviewer's concern.
> > >
> > > Q4 / W4: comparison.
> > >
> > > A4:
> > >
> > > We thank the reviewer for the continued feedback.
> > >
> > > As noted in our first-round response, the compared methods were originally designed for different purposes and application settings; it is objectively difficult to align them perfectly across all dimensions.
> > >
> > > Moreover, Table 3 provides an apples-to-apples comparison under nearly identical conditions (backbone, scale, training regime, compute). Our "two-stage design + AFS + reward" achieves the best results.
> > >
> > > To further address this, we retrained Seg-R1-7B using nearly the same training data, model scale (Qwen2.5-VL-7B), and compute resources (4×A800) as ours:
> > >
> > > |Method|Ana.M|Ana.C|Ans.M|Ans.C|cloU|
> > > |-|-|-|-|-|-|
> > > |Seg-R1(retrained)|0.323|0.305|0.849|3.819|58.28|
> > > |Ours|0.541|1.520|0.933|6.610|62.71|
> > >
> > > These results provide stronger apples-to-apples evidence. Even under identical conditions, the retrained Seg-R1 remains substantially behind our method. This confirms that the performance gain comes from our methodological advantages, not training configurations.

---

### Decision · Program_Chairs · 2026-04-30

**Decision:**

Accept (regular)

**Comment:**

The paper proposes EARL, an efficient framework for egocentric interaction reasoning and pixel grounding, built on a two-stage design incorporating the AFS module, GPRO-based RL, and other techniques. Reviewers XBSG, UMca, and PuYT provided positive assessments, highlighting the solid overall framework, comprehensive experimental evaluation, and clear writing. They also agreed that the paper addresses a meaningful problem in egocentric vision with strong practical relevance. While the reviewers have concerns regarding the fundamental novelty, they nevertheless appreciate the solidity of the overall approach and the strong in-domain and out-of-domain performance gains. After considering the reviews and the authors’ response, the ACs recommend accepting this paper.